# Screening for Small Molecule Modulators of *Trypanosoma brucei* Hsp70 Chaperone Activity Based upon Alcyonarian Coral-Derived Natural Products

**DOI:** 10.3390/md18020081

**Published:** 2020-01-27

**Authors:** Sarah K. Andreassend, Stephen J. Bentley, Gregory L. Blatch, Aileen Boshoff, Robert A. Keyzers

**Affiliations:** 1School of Chemical and Physical Sciences, Victoria University of Wellington, Wellington 6012, New Zealand; sarah.andreassend@vuw.ac.nz; 2Centre for Biodiscovery, Victoria University of Wellington, Wellington 6012, New Zealand; 3Biotechnology Innovation Centre, Rhodes University, Grahamstown 6140, South Africa; stephenjohnbentley@gmail.com (S.J.B.); aileenboshoff18@gmail.com (A.B.); 4Biomedical Biotechnology Research Unit, Department of Biochemistry and Microbiology, Grahamstown 6140, South Africa; g.blatch@ru.ac.za; 5Office of the PVC Research, The University of Notre Dame Australia, Fremantle, Western Australia 6959, Australia

**Keywords:** anti-parasitic, heat shock protein, malonganenone, SAR, African Trypanosomiasis

## Abstract

The *Trypanosoma brucei* Hsp70/J-protein machinery plays an essential role in survival, differentiation, and pathogenesis of the protozoan parasite, and is an emerging target against African Trypanosomiasis. This study evaluated a set of small molecules, inspired by the malonganenones and nuttingins, as modulators of the chaperone activity of the cytosolic heat inducible *T. brucei* Hsp70 and constitutive TbHsp70.4 proteins. The compounds were assessed for cytotoxicity on both the bloodstream form of *T. b. brucei* parasites and a mammalian cell line. The compounds were then investigated for their modulatory effect on the aggregation suppression and ATPase activities of the TbHsp70 proteins. A structure–activity relationship for the malonganenone-class of alkaloids is proposed based upon these results.

## 1. Introduction

The etiological agent of African Trypanosomiasis, *Trypanosoma brucei* (*T. brucei*), is an extracellular blood- and tissue-borne unicellular parasitic protozoan. It gives rise to infection in both humans and animals, predominantly across sub-Saharan Africa, and is transmitted to its mammalian host during a blood meal of the infected tsetse fly vector (*Glossina* spp.), which ensures the cyclical transmission of the parasite between numerous hosts [1]. There is a dire need for the development of more effective and safer drugs to treat the disease, because of the toxicity and long duration for the current treatments, coupled to the increase of drug resistance in trypanosomes and the lack of a vaccine [2,3]. Molecular chaperones have been shown to play an essential role in stress-induced stage differentiation and are vital for disease progression and transmission [4,5], making this protein family an attractive anti-parasitic chemotherapeutic target.

The highly ubiquitous 70-kDa heat shock protein (Hsp70) family of molecular chaperones, known as HSPA in humans, is one of the most evolutionarily conserved protein families. It is involved in a plethora of essential cellular functions that include promoting the correct protein folding of newly synthesized polypeptides, mediating protein translocation, and facilitating proteolytic degradation of non-native and aggregated proteins [6,7]. The domain architecture of eukaryotic cytosolic Hsp70s is typically comprised of an N-terminal nucleotide binding domain connected via a linker region to a C-terminal domain with a substrate binding domain, and a 10-kDa α-helical domain with a conserved EEVD motif [8,9]. The function and specificity of Hsp70s are regulated by the 40-kDa heat shock protein (Hsp40) family, also referred to as J-proteins, due to the presence of their signature domain, the conserved ~70 amino acid region known as the J-domain [10], which interacts with the nucleotide binding domain of Hsp70. J-proteins function as a co-chaperone of Hsp70 by delivering specific substrates and stimulating the low intrinsic ATPase activity of Hsp70 [10]. J-proteins are classified into four types, with types I and II binding protein and preventing aggregation of unfolded proteins, thereby displaying a holding-function [11].

The *Trypanosoma brucei* Hsp70 (TbHsp70) and J-protein families have undergone greater evolutionary expansion relative to other eukaryotic systems, and contain diverse family members [12]. RNAi-mediated knockdown of *T. brucei* genes conducted by Alsford and colleagues [13] demonstrated that the Hsp70/J-protein machinery plays a prominent role in trypanosome biology, as the loss of certain members of these protein families was found to be lethal at one or more stages in its life cycle. It has been proposed that TbHsp70 plays an essential role in cytoprotection during cellular stress [12], and *in vivo* studies on the Type I cytosolic J-protein, Tbj2, have shown that it is stress inducible and essential [14]. Furthermore, evidence from *in vitro* assays [15] suggested that Tbj2 has chaperone (e.g., able to suppress protein aggregation of model substrates) and co-chaperone properties (e.g., able to stimulate the ATPase activity of a trypanosomal Hsp70).

Several promising studies have been conducted on assessing the potential of naturally occurring marine- or plant-based extractables as modulators of the Hsp70 chaperone system in *Plasmodium falciparum (P. falciparum)* [16,17,18,19,20]. Cockburn and colleagues [18] investigated a set of small molecules derived from two classes of compounds, 1,4-naphthoquinones and marine prenylated alkaloids, for modulation of the activity of two biologically important plasmodial Hsp70s. One of the compounds, malonganenone A, showed desirable properties as a plasmodial Hsp70 modulator, as the compound inhibited the steady-state and J-protein stimulated ATPase activity of plasmodial Hsp70s, and not that of human Hsp70 [18]. It was also shown to disrupt the interaction between the exported PfHsp70-x and J-protein, marking malonganenone A for further study particularly with the synthesis of analogues that have more potent antimalarial activities and higher selectivity as PfHsp70 inhibitors [18].

The malonganenones are a family of tetraprenylated alkaloid marine natural products isolated from gorgonian sea fans, collected in Africa and China [21,22,23,24]. To date, a total of 17 malonganenones (A–Q) have been isolated, as well as six closely related nuttingins (A–F) (Figure 1). The malonganenones and nuttingins are cytotoxic against several cancer cell lines (IC_50_ 0.35–84.9 μM) [21,23] while malonganenones D–H and nuttingins A–F cause apoptosis of transformed mammalian cells (1.25 μg/mL) [22]. Additionally, malonganenones L and Q are inhibitory against phosphodiesterase-4D (IC_50_ 8.5 and 20.3 μM) [24] and malonganenone D reduces c-Met kinase activity 2-fold (10 μM) [23]. Importantly, malonganenones A and C are anti-plasmodial against *P. falciparum* (IC_50_ 0.81 and 5.20 μM) [17].

The malonganenones vary mainly in the composition of the nitrogenous head group, with small changes at the end of the prenyl side chain. Bioactivity mainly varies relative to the identity of the head group, suggesting that they play the primary role of pharmacophore. Therefore, a simpler prenyl chain, as in malonganenone J, could be substituted for the other natural product side chains, and still be expected to present useful bioactivity. Although the side chain may be less important for bioactivity, it still plays a significant role. A simplified analogue of malonganenone B, which substituted the side chain with a methyl group, was completely inactive in the same assay against PfHsp70-1, whereas the original compound’s activity was dose-dependent [17]. This result suggests that an extended side chain could be important for adding lipophilicity. Therefore, in this study, a structure–activity relationship (SAR) of the side chain length was probed by varying the length from one to three prenyl units. Analogues of the malonganenone and nuttingins were synthesized by alkylation of *N*-heterocyclic compounds, while analogues of malonganenone C were synthesized by simple derivatization of terpenoid amines. It should be noted that malonganenone J is the only member of the family to have succumbed to total synthesis to date [25].

Overall, this study aimed to evaluate potential inhibitors, inspired by the malonganenones and nuttingins, of the chaperone activity of *T. brucei* Hsp70 proteins. The compounds were assessed for cytotoxicity on both the bloodstream form of *T. b. brucei* parasites and a mammalian cell line. The compounds were then investigated for their modulatory effect on the aggregation suppression and ATPase activities of cytosolic TbHsp70 (homologue of the cytosolic inducible human Hsp70, HSPA1A) and TbHsp70.4 (homologue of the constitutive human Hsp70, HSPA8). The human chaperone HSPA8 and its co-chaperone DNAJB2 (Hsj1a), a Type II J-protein, were chosen as model representatives for investigation of the modulatory effect of the small molecules on a human Hsp70/J-protein partnership. HSPA8 has been shown to be involved in essential housekeeping functions [26,27], as knockout of the gene in mice was shown to be lethal [28], and RNAi-mediated knockdown resulted in massive cell death in various cell types [29]. DNAJB2 has been shown to be preferentially expressed in neuronal cells [30], where it plays a prominent role in protein degradation [31], and it has been shown to stimulate the *in vitro* basal ATPase activity of HSPA8 [32], HSPA1A [16,18,33], PfHsp70-1 [16,18], and PfHsp70-x [18]. The intention was to identify inhibitors that were specific to modulating the activities of the trypanosomal Hsp70s, TbHsp70 and/or TbHsp70.4. Overall, this study aimed to identify a potential new avenue to African trypanosomiasis chemotherapy.

## 2. Results and Discussion

### 2.1. Synthesis of Malonganenone and Nuttingin Analogues

Coverage of chemical space was maximized for the first generation of malonganenone and nuttingin analogues by using non-selective alkylation conditions and terpenoid bromides, with geometric mixtures at the C-2 alkene, to give maximal compound diversity. These conditions tended to efficiently yield multiple regio- and geometric isomers per reaction. The regioisomers were easily separated by chromatography, but the geometric isomers proved to be more difficult. The (2*E*)*-* and (2*Z*)-isomers of the farnesyl derivatives were often isolated as mixtures and were tested as such. The geometric isomers of the geranylgeranyl derivatives were markedly easier to separate by chromatography.

The purines, adenine (**1**), 6-(dimethylamino) purine (**2**), and 6-chloropurine (**3**) were alkylated with prenyl bromides (Scheme 1). Alkylation of **1** yielded *N*-3- (**4**, **6**) and *N*-9 (**7**, **8**) monoalkylated products. Alkylation with geranyl bromide produced a minor amount of a dialkylated species (**5**), which was not isolated from the other reactions. Alkylation of **2** also yielded *N*-3- (**10**, **12**) and *N*-9 (**9**, **11**, **13**, **14**) monoalkylated regioisomers. The ^1^H NMR spectra of **9–14** had broad *N*-methyl resonances, which were only equivalent for the *N*-9 isomers. This was rationalized by the formation of a stable imine resonance form for the *N*-3 regioisomer, which has also been observed previously for a similar *N*-3 alkylated 6-(dimethylamino)purine [34]. Alkylation of **3** yielded *N*-9- (**15**, **17**, **19**), and *N*-7 (**16**, **18**, **20**, **21**) monoalkylated regioisomers.

Head groups based on purinones were also alkylated since the head group of nine malonganenones comprise a hypoxanthine core, with a carbonyl at C-6. Xanthine (**22**), 3-methylxanthine (**23**), and theophylline (**24**) are related by increasing methylation; theophylline is the head group of nuttingin A and B. The other two purinones selected were 2-mercapto-3-methylhypoxanthine (**25**) and 1,3-dimethyluric acid (**26**). Alkylation of the methyl series of purinones, and **25**, all yielded *N*-7 monoalkylated regioisomers (**29**–**41**), except for **22**, which only yielded *N*-3,*N*-7 dialkylated species (**27**, **28**) (Scheme 2).

Alkylation of **26** yielded *C*-5 monoalkylated regioisomers (**42**, **43**) and only a minor amount of *N*-9 monoalkylated regioisomer (**44**) from the farnesyl bromide reaction (Scheme 3). We suggest that these species result from enolate reactivity, providing a simple method for forming quaternary carbon-carbon bonds (Scheme 4).

Alkylation of **45** yielded *N*-5 monoalkylated analogues (**46**, **49**), as well as *N*-1, *N*-5- (**47**, **50**) and *N*-2, *N*-7 (**48**) dialkylated analogues, presumably from enhanced nucleophilicity of the pyrimidine ring (Scheme 5).

Other readily available *N*-heterocyclic compounds were also alkylated, including the pyrimidines, uracil (**51**), thymine (**52**), and cytosine (**53**). Alkylation of **51**, and **52** yielded only *N*-1 monoalkylated derivatives (**54**–**59**) whereas **53** only produced *N*-1,*N*-3 dialkylated products (**61**, **63**) (Scheme 6). In addition to the dialkylated cytosines, formylated imine derivatives (**60**, **62**, **64**) were also isolated, likely formed via the reaction solvent, DMF. Although formylation with DMF usually requires pre-activation, such as in the Vilsmeier–Haack reaction [35], milder conditions have also been successful at yielding formylated derivatives. For example, moderate heating of DMF and imidazole formylated a variety of amino acids and primary amines [36]. The proposed mechanism suggests that imidazole acts as an intermediate acyl transfer reagent (Scheme 7), therefore it is feasible that cytosine could also fulfil this role. The formyl imidazole produced is itself further reactive when other amine nucleophiles are present. The absence of stronger nucleophiles in these alkylation reactions resulted in the isolation of formyl cytosine.

Alkylation of potassium phthalimide (**65**) (Scheme 8) and imidazole (**66**) (Scheme 9) furnished the last of the *N*-heterocyclic analogues. Both reactions proceeded straightforwardly to yield *N*-1 monoalkylated analogues (**67**–**72**).

Malonganenones C, H, and K, have a simple formamide head group and analogues were synthesized by formylation [37,38] or acetylation [39,40] of prenyl amines [41], followed by methylation [42] (Scheme 10). The ^1^H NMR spectra of the formamide- (**73**–**77**) and *N*-methyl formamide (**78**–**80**) series suggested a mixture of two rotamers, with doubling of most resonances near the head group. These observations are consistent with previous data reported for geranyl formamide [43] and of more relevance, malonganenone B [37]. Conversely, the acetamide analogues (**81**–**85**) were not rotameric but became so upon *N*-methylation (**86**–**88**), analogous to DMF.

### 2.2. Evaluation of the Potential Small Molecule Inhibitors for Cytotoxicity on Mammalian Cells and Parasites

A resazurin-based cytotoxicity assay was conducted to assess the potential anti-trypanosomal activity of the 74 compounds (Appendix A). Seven were identified to be non-toxic as the parasite survival was ≥100% (Appendix A).

The compounds were further assessed for toxicity using HeLa (human cervix adenocarcinoma) cells. Emetine is a natural alkaloid that has been shown to induce apoptosis in mammalian cell lines [44] and was incorporated into the study as a positive control, displaying high toxicity at 10 μM towards HeLa cells (6.03% cell viability). Compounds **5**, **61**, and **62** were toxic to HeLa cells at 20 μM (Appendix A). Comparison of the toxicity of the compounds at 20 μM toward both trypanosomes and HeLa cells indicates that the toxicity of most compounds is selective towards the parasite (Appendix A).

Most of the synthetic compounds were shown to display high anti-trypanosomal activity; 48 compounds were selected for further assessment as they reduced parasite growth ≥80% at 20 µM (Appendix A). Dose–response curves of the 48 selected compounds were generated, and the IC_50_ (50% inhibitory concentration) values for each compound was determined (Table 1, Appendix A). For comparative purposes, pentamidine, an existing drug used to treat the early stages of Human African trypanosomiasis caused by *T. b. gambiense* [45,46] was used as the positive control.

Although none of the tested compounds are comparable to the drug standard pentamidine in potency, tentative SAR can be proposed from the results to aid in designing a second generation of analogues. SAR analysis of the side chain suggests that length correlates positively with activity as most of the shortest side chain derivatives are inactive and activity increases upon lengthening the side chain from farnesyl to geranylgeranyl (i.e., **7** vs. **8**, **55** vs. **56**, **68** vs. **69**, **71** vs. **72**). The activity is further influenced by the head group. Activating head groups have shorter optimal chain lengths and further lengthening has minimal effect (i.e., **11** vs. **14**, **18** vs. **20**, **21**) whereas the deactivating head group series are completely inactive (i.e., **38**–**41**), or activity is only rescued at the longest side chain length (i.e., **59**, **37**). No general trends can be derived from the identity of the head group, but small changes such as methylation seem to have a large effect on activity. For example, 3-methylxanthine (**23**) and theophylline (**24**) differ by the presence of an *N*-1 methyl and while all side chain lengths of the *N*-7 alkylated 3-methylxanthines (**29–33**) are active, only the *N*-7 geranylgeranyl theophyllines (**36**, **37**) display activity. Converse to *N*-methyl deactivation, the acetamides are all more active than the formamides by 5- to 13-fold (**74** vs. **82**, **76** vs. **84**, **77** vs. **85**).

The SAR pertinent to alkene geometry generally suggests that the (2*Z*)-isomers or (2*Z*/*E*)-mixtures are more active than the (2*E*)-isomers, where the latter is inactive (i.e., **74** vs. **75**) or activity is reduced by 1.5- to 5-fold (i.e., **36** vs. **37**, **82** vs. **83**, **84** vs. **85**).

Of the 48 compounds screened, 22 compounds were shown to display high toxicity towards parasites with IC_50_ values all below 10 µM (Table 1, Appendix A). However, the positive control pentamidine was shown to display the highest toxicity towards the parasite (IC_50_ 5.3 nM). Despite this, pentamidine is limited to treating the haemolymphatic stage of *T. b. gambiense* infections [47], and pentamidine resistance has been reported [48]. From these 22 compounds, the top five active compounds (**28**, **47**, **31**, **27**, and **63**), and two further compounds (**60** and **48**) with abundantly available mass, were selected for further assessment. Even though the mechanism of inhibition of the parasites by the selected compounds is currently unknown, their modulatory effect on the molecular chaperone activity of TbHsp70 proteins was evaluated.

### 2.3. Modulation of the Aggregation Suppression Activity of TbHsp70 and TbHsp70.4

The malate dehydrogenase (MDH) aggregation suppression assay was used to evaluate the modulatory effect of the seven selected compounds, that were highly toxic towards the parasites with IC_50_ values all below 7 μM (**27**, **47, 48**, **31**, **28**, **60**, and **63**), on the chaperone activities of TbHsp70 and TbHsp70.4. Human HSPA8 was not suitable for MDH aggregation suppression experiments, as the protein is prone to aggregation at 48 °C (Appendix A). The target compounds (in the absence of the Hsp70) were shown to have no effect on the aggregation of MDH (Appendix A), and thus were ruled out as chemical chaperones. DMSO (1% *v*/*v*) had no significant effect on the chaperone activity of both *T. brucei* Hsp70s (not shown). Increasing concentrations of the compounds resulted in a general increase in % MDH aggregation due to inhibition of the aggregation suppression activity of TbHsp70 and TbHsp70.4 (Figure 2A,B). Peptide substrate motifs recognized by Hsp70s have been described as having a central hydrophobic region of four to five residues, flanked by basic residues [49]. It is possible that the hydrophobic hydrocarbon side chains of the compounds are binding to the hydrophobic pocket of the substrate binding domains of the *T. brucei* Hsp70s, preventing MDH from binding.

### 2.4. The Basal ATPase Activities of TbHsp70, TbHsp70.4 and HsHSPA8 Can Be Differentially Modulated

An initial screen of the modulatory effects of the selected compounds (**27**, **47, 48**, **31**, **28**, **60**, and **63**) on the basal ATPase activity of the TbHsp70s was conducted at single concentrations (300 μM) (Appendix A). DMSO (1% *v*/*v*) was also shown to have no significant effect on the ATPase activity of both *T. brucei* Hsp70s and HsHSPA8 (Appendix A). The compounds inhibited the basal ATPase activities of TbHsp70 and TbHsp70.4 to different extents. In comparison to TbHsp70.4, **31**, **28**, and **60** inhibited the basal ATPase activity of TbHsp70 to a lesser extent, with **60** being the least effective with 39% inhibition (Appendix A). These molecules may have a selective mode of inhibition through interaction with specific residues of the targeted domain in TbHsp70.4. Compounds which reduced the basal ATPase activities of the TbHsp70s by ≥70% at 300 μM (Appendix A) were further tested over a range of concentrations (Figure 3). The selected compounds inhibited the basal ATPase activities of TbHsp70.4 and TbHsp70 in dose-dependent manners (Figure 3). Compound **47** inhibited the ATPase activities to the greatest extent for both TbHsp70s, with the lowest concentration of 1 μM inhibiting the basal ATPase activity of TbHsp70.4 by 66% and TbHsp70 by 44%, respectively. Due to limited availability, only compounds **27**, **28**, **31**, **47**, and **63** were further assessed for inhibition of the basal ATPase activity of HsHSPA8 over a range of concentrations. The degree of inhibition of the basal HsHSPA8 ATPase activity by the same compounds was reduced in comparison to the TbHsp70s (Figure 3). The contrast in inhibition between the trypanosomal and human Hsp70 proteins may be due to a greater binding affinity of the compounds to the *T. brucei* Hsp70s than to human HSPA8.

### 2.5. Modulation of the J-Protein-Stimulated ATPase Activities of TbHsp70, TbHsp70.4 and HsHSPA8

Due to the promising results obtained, the remaining available compounds (**27**, **28**, **31**, **47**, and **63**) were investigated for modulation of J-stimulated ATPase activity of the *T. brucei* Hsp70s, and possible disruption of the Hsp70/J-protein partnerships. A preliminary screen of the modulatory effects of the five compounds at 300 μM was performed on the Tbj2-stimulated ATPase activities of TbHsp70 and TbHsp704 and the HsDnaJB2-stimulated ATPase activity of HsHSPA8 (Appendix A). DMSO (1% *v*/*v*) was also shown to have no significant effect on the J-stimulated ATPase activity of both *T. brucei* Hsp70s and HsHSPA8 (Appendix A). Tbj2 stimulated the ATPase activities of TbHsp70 3.23-fold and TbHsp70.4 2.98-fold, whilst HsDNAJB2 stimulated the ATPase activity of HsHSPA8 ~5-fold (data not shown). All the tested compounds were shown to inhibit the J-stimulated ATPase activities of TbHsp70 and TbHsp70.4 by ~60% and inhibit the J-stimulated ATPase activity of HsHSPA8 by ~40% (Appendix A) The addition of increasing concentrations of the compounds (**27**, **28**, **31**, **47**, and **63**) reduced the J-stimulated ATPase activities of TbHsp70 and TbHsp70.4, with the greatest reduction observed for TbHsp70 (Figure 4). The compounds tested were also shown to suppress the DNAJB2-stimulated ATPase activity of HsHSPA8 (Figure 4). The small molecules were shown to have a decreased effect on the J-stimulated ATPase activity of HsHSPA8 in comparison to the TbHsp70s (Figure 4), though the results indicate that Tbj2 is stimulating the compound-induced inhibition of the basal ATPase activity.

An additional experiment was carried out to assess the effect of varying the order of addition of the compounds (**27**, **28**, **31**, **47**, and **63**) at single concentrations (150 μM) and Tbj2 in the assay. This was conducted to elucidate if the small molecules were disrupting the Hsp70/J-protein partnership. However, no significant difference was observed in the inhibition of the J-stimulated ATPase activity for both *T. brucei* Hsp70s by varying the order of addition of the reaction components (*p* < 0.05; Figure 5). This suggests that the small molecules are not disrupting the binding of Tbj2 to the TbHsp70s but are binding to a site, independent of the Tbj2 binding site.

## 3. Materials and Methods

### 3.1. General Synthetic Procedures

All reactions were carried out under an inert atmosphere (Ar or N_2_), with oven or vacuum-dried glassware, using standard syringe techniques and dry solvents. Dry DCM, Et_2_O, and THF were obtained from a Puresolv. system (Innovative Technology). Triethylamine was distilled from CaH_2_. Methyl iodide was run through a plug of activated alumina prior to use. 2-Mercapto-3-methylhypoxanthine [50], the relevant terpenoid bromides [51,52,53], and terpenoid amines [41] were synthesized according to literature. The C-2 alkene *E*/*Z* ratio of the terpenoid bromides was determined from integrated peak areas detected by ^1^H NMR spectroscopy. Geranyl bromide was entirely the (2*E*)-isomer as set by the (2*E*)-configuration of the geraniol starting material. Farnesyl bromide was synthesized from an isomeric mixture of farnesol, with an *E*/*Z* ratio of 3:2. Geranylgeranyl bromide was synthesized from racemic geranyllinalool, with the resulting *E*/*Z* ratio of 3:1. All other solvents and reagents were used as received from commercial suppliers. All compounds were purified by silica gel flash chromatography, using silica gel 60 (40–63 micron), unless otherwise stated. Thin-layer chromatography was performed on Macherey-Nagel, POLYGRAM^®^ Sil G/UV254 plates, and were visualized with a UV lamp, iodine-, ceric ammonium molybdate-, vanillin-, or sulfuric acid stain. 1D (^1^H, ^13^C, NOESY) and 2D (COSY, HSQC, HMBC) NMR spectra were recorded using Varian Inova or DirectDrive instruments operating at 300 (Inova), 500 (Inova), or 600 (DirectDrive) MHz for proton and 125 or 150 MHz for carbon. IR spectra were obtained using an ALPHA FT-IR spectrometer (Bruker). MS data and tandem MS data were obtained using an Agilent 6530 Q-TOF LC/MS high-resolution mass spectrometer equipped with an Agilent 1260 HPLC system for sample introduction.

### 3.2. General Alkylation Procedure for Synthesis of ***4–21**, **27–50**, **54–64**, **67–72***

#### 3.2.1. (*E*)-3-(3,7-Dimethylocta-2,6-dien-1-yl)-3*H*-purin-6-amine (**4**) and 3,9-bis((*E*)-3,7-dimethylocta-2,6-dien-1-yl)-3,9-dihydro-6*H*-purin-6-imine (**5**)

Adenine (1.00 mmol, 134.7 mg) and K_2_CO_3_ (1.31 mmol, 181.3 mg) were stirred in DMF (2 mL) at RT for 10 min before dropwise addition of geranyl bromide (1.2 mmol, 250.6 mg). The reaction was stirred for 21 h, then poured onto H_2_O (6 mL) and extracted with EA (3 × 2 mL). The combined extracts were washed with H_2_O (3 × 2 mL), then brine (1 × 2 mL) and dried over anhydrous MgSO_4_. The dried residue was purified by silica gel flash chromatography (5% MeOH/EA) to yield **4** and **5**.

Compound **4**: 51.9 mg (19%), pale-yellow crystals; R*_f_* = 0.80 (5% MeOH/EA); ^1^H NMR (500 MHz, CDCl_3_): δ 8.06 (s, 1H, H-8), 8.02 (s, 1H, H-2), 5.49 (t, *J* = 7.3 Hz, 1H, CH=), 5.07–5.03 (m, 1H, CH=), 5.01 (d, *J* = 7.3 Hz, 2H, NCH_2_), 2.12 (br s, 4H, 2 × CH_2_), 1.83 (s, 3H, CH_3_), 1.66 (s, 3H, CH_3_), 1.57 (s, 3H, CH_3_); ^13^C{^1^H} NMR (150 MHz, CDCl_3_): δ 154.3 (C, C-6), 153.9 (CH, C-8), 150.9 (C, C-4), 145.0 (C=), 141.7 (CH, C-2), 132.5 (C=), 123.4 (CH=), 120.8 (C, C-5), 116.2 (CH=), 47.4 (NCH_2_), 39.6 (CH_2_), 26.2 (CH_2_), 25.8 (CH_3_), 17.9 (CH_3_), 16.8 (CH_3_); IR (film from CH_2_Cl_2_): ν_max_ 3231, 3067, 2966, 2912, 2853 cm^−1^; HRESIMS *m*/*z*: [M + H]^+^ Calcd. for C_15_H_22_N_5_ 272.1870; Found 272.1875 (Δ = 1.8 ppm); HRESIMS/MS (40 eV) *m*/*z* (%): 136.0612 (100), 81.0700 (17).

Compound **5**: 7.8 mg (3%), pale-yellow oil; R*_f_* = 0.04 (10% MeOH/EA); ^1^H NMR (600 MHz, CDCl_3_): δ 7.68 (s, 1H, H-8), 7.26 (s, 1H, H-2), 5.47 (t, *J* = 7.3 Hz, 1H, CH=), 5.35 (t, *J* = 7.6 Hz, 1H, CH=), 5.15 (d, *J* = 7.3 Hz, 2H, NCH_2_), 5.08–5.00 (m, 2H, 2 × CH=), 4.60 (d, *J* = 7.1 Hz, 2H, NCH_2_), 2.15–2.03 (m, 8H, 4 × CH_2_), 1.78 (s, 3H, CH_3_), 1.77 (s, 3H, CH_3_), 1.67 (s, 3H, CH_3_), 1.65 (s, 3H, CH_3_), 1.58 (s, 3H, CH_3_), 1.57 (s, 3H, CH_3_); ^13^C{^1^H} NMR (150 MHz, CDCl_3_): δ 155.2 (C, C-6), 144.9 (CH, C-8), 144.6 (C, C-4), 143.5 (C=), 143.0 (C=), 140.5 (CH, C-2), 132.43 (C=), 132.38 (C=), 123.7 (CH=), 123.4 (CH=), 117.0 (CH=), 116.4 (CH=), 112.6 (C, C-5), 45.7 (NCH_2_), 46.1 (NCH_2_), 39.63 (CH_2_), 39.60 (CH_2_), 26.23 (CH_2_), 26.22 (CH_2_), 25.9 (CH_3_), 25.8 (CH_3_), 17.88 (CH_3_), 17.85 (CH_3_), 16.79 (CH_3_), 16.76 (CH_3_); IR (film from CH_2_Cl_2_): ν_max_ 2966, 2916, 2855, 1629 cm^−1^; HRESIMS *m*/*z*: [M + H]^+^ Calcd. for C_25_H_38_N_5_ 408.3122; Found 408.3129 (Δ = 1.7 ppm); HRESIMS/MS (20 eV) *m*/*z* (%): 272.1851 (41), 136.0608 (100), 81.0698 (20).

#### 3.2.2. 3-((6*E*)-3,7,11-Trimethyldodeca-2,6,10-trien-1-yl)-3*H*-purin-6-amine (**6**) and 9-((6*E*)-3,7,11-trimethyldodeca-2,6,10-trien-1-yl)-9*H*-purin-6-amine (**7**)

Adenine (0.51 mmol, 68.3 mg), K_2_CO_3_ (0.52 mmol, 71.9 mg) and farnesyl bromide (0.55 mmol, 157.0 mg) in DMF (2 mL) at 50 °C for 27 h yielded **6** and **7**, with modified work up-H_2_O (6 mL) was added to the reaction filtrate and stored in the fridge until precipitate formed. The isolated solid was purified by chromatography.

Compound **6**: 22.4 mg (13%), pale-yellow crystals; R*_f_* = 0.17 (5% MeOH/EA); 3:2 *E*/*Z*, data for major isomer: ^1^H NMR (600 MHz, CDCl_3_): δ 8.05 (s, 1H, H-8), 8.00 (s, 1H, H-2), 5.51–5.45 (m, 1H, CH=), 5.10–5.03 (m, 2H, 2 × CH=), 5.01 (d, *J* = 7.3 Hz, 2H, NCH_2_), 2.15–2.09 (m, 4H, 2 × CH_2_), 2.05–1.98 (m, 2H, CH_2_), 1.98–1.92 (m, 2H, CH_2_), 1.83 (s, 3H, CH_3_), 1.65 (s, 3H, CH_3_), 1.57 (s, 6H, 2 × CH_3_); ^13^C{^1^H} NMR (150 MHz, CDCl_3_): δ 154.5 (C, C-6), 154.1 (CH, C-8), 150.9 (C, C-4), 144.9 (C=), 141.6 (CH, C-2), 136.1 (C=), 131.5 (C=), 124.3 (CH=), 123.3 (CH=), 121.1 (C, C-5), 116.2 (CH=), 47.4 (NCH_2_), 39.8 (CH_2_), 39.6 (CH_2_), 26.8 (CH_2_), 26.2 (CH_2_), 25.8 (CH_3_), 17.8 (CH_3_), 16.9 (CH_3_), 16.2 (CH_3_); IR (film from CH_2_Cl_2_): ν_max_ 3231, 3065, 2965, 2915, 2855, 1704 cm^−1^; HRESIMS *m*/*z*: [M + H]^+^ Calcd. for C_20_H_30_N_5_ 340.2496; Found 340.2505 (Δ = 2.6 ppm); HRESIMS/MS (40 eV) *m*/*z* (%): 136.0610 (100), 119.0345 (13).

Compound **7**: 3.6 mg (2%), white powder; R*_f_* = 0.30 (EA); 3:2 *E*/*Z*, NMR data for major isomer: ^1^H NMR (500 MHz, CDCl_3_): δ 8.38 (s, 1H, H-2), 7.78 (s, 1H, H-8), 5.59 (br s, 2H, NH_2_), 5.45 (t, *J* = 7.1 Hz, 1H, CH=), 5.13–5.03 (m, 2H, 2 × CH=), 4.78 (d, *J* = 7.1 Hz, 2H, NCH_2_), 2.19–2.08 (m, 4H, 2 × CH_2_), 2.08–1.92 (m, 4H, 2 × CH_2_), 1.81 (s, 3H, CH_3_), 1.67 (s, 6H, 2 × CH_3_), 1.59 (s, 3H, CH_3_); ^13^C{^1^H} NMR (150 MHz, CDCl_3_): δ 155.4 (C, C-6), 153.0 (CH, C-2), 150.1 (C, C-4), 143.0 (C=), 140.2 (CH, C-8), 135.9 (C=), 131.6 (C=), 124.3 (CH=), 123.5 (CH=), 119.7 (C, C-5), 117.5 (CH=), 41.4 (NCH_2_), 39.8 (CH_2_), 39.6 (CH_2_), 26.8 (CH_2_), 26.3 (CH_2_), 25.9 (CH_2_), 17.9 (CH_3_), 16.7 (CH_3_), 16.2 (CH_3_); IR (film from CH_2_Cl_2_): ν_max_ 3307, 3140, 2965, 2924, 2865, 1601 cm^−1^; HRESIMS *m*/*z*: [M + H]^+^ Calcd. for C_20_H_30_N_5_ 340.2496; Found 340.2501 (Δ = 1.5 ppm); HRESIMS/MS (40 eV) *m*/*z* (%): 136.0613 (100), 119.0346 (12).

#### 3.2.3. 9-((2*E*,6*E*,10*E*)-3,7,11,15-Tetramethylhexadeca-2,6,10,14-tetraen-1-yl)-9*H*-purin-6-amine (**8**)

Adenine (0.69 mmol, 93.8 mg), K_2_CO_3_ (0.78 mmol, 108.0 mg) and geranylgeranyl bromide (0.73 mmol, 259 mg) in DMF (1 mL) for 27 h yielded **8**, 9.9 mg (4%), pale-yellow crystals. R*_f_* = 0.32 (EA); ^1^H NMR (500 MHz, CDCl_3_): δ 8.37 (s, 1H, H-2), 7.77 (s, 1H, H-8), 5.77 (br s, 2H, NH_2_), 5.44 (t, *J* = 7.2 Hz, 1H, CH=), 5.11–5.05 (m, 3H, 3 × CH=), 4.77 (d, *J* = 7.2 Hz, 2H, NCH_2_), 2.16–2.08 (m, 4H, 2 × CH_2_), 2.08–2.01 (m, 4H, 2 × CH_2_), 2.01–1.92 (m, 4H, 2 × CH_2_), 1.81 (s, 3H, CH_3_), 1.67 (s, 3H, CH_3_), 1.59 (s, 9H, 3 × CH_3_); ^13^C{^1^H} NMR (150 MHz, CDCl_3_): δ 155.5 (C, C-6), 153.0 (CH, C-2), 150.1 (C, C-4), 143.0 (C=), 140.1 (CH, C-8), 135.9 (C=), 135.2 (C=), 131.4 (C=), 124.5 (CH=), 124.2 (CH=), 123.5 (CH=), 119.7 (C, C-5), 117.5 (CH=), 41.3 (NCH_2_), 39.9 (CH_2_), 39.8 (CH_2_), 39.6 (CH_2_), 26.9 (CH_2_), 26.7 (CH_2_), 26.3 (CH_2_), 25.8 (CH_3_), 17.8 (CH_3_), 16.7 (CH_3_), 16.2 (CH_3_), 16.1 (CH_3_); IR (film from CH_2_Cl_2_): ν_max_ 3468, 3324, 3153, 3051, 2969 2918 cm^−1^; HRESIMS *m*/*z*: [M + H]^+^ Calcd. for C_25_H_38_N_5_ 408.3122; Found 408.3129 (Δ = 1.7 ppm); HRESIMS/MS (40 eV) *m*/*z* (%): 136.0614 (100).

#### 3.2.4. (*E*)-9-(3,7-Dimethylocta-2,6-dien-1-yl)-*N*,*N*-dimethyl-9*H*-purin-6-amine (**9**) and (*E*)-3-(3,7-dimethylocta-2,6-dien-1-yl)-*N*,*N*-dimethyl-3*H*-purin-6-amine (**10**)

6-(Dimethylamino)purine (0.17 mmol, 28.3 mg), Na_2_CO_3_/K_2_CO_3_ (1:1, 108 mg) and geranyl bromide (0.42 mmol, 90.8 mg) in DMF (3 mL) for 48 h yielded **9** and **10**, with modified work up–concentration under reduced pressure.

Compound **9**: 9.2 mg (18%), white powder; R*_f_* = 0.12 (1:2 EA/PE); ^1^H NMR (600 MHz, CDCl_3_): δ 8.35 (s, 1H, H-2), 7.69 (s, 1H, H-8), 5.43 (t, *J* = 7.1 Hz, 1H, CH=), 5.05 (t, *J* = 6.1 Hz, 1H, CH=), 4.75 (d, *J* = 7.1 Hz, 2H, NCH_2_), 3.53 (br s, 6H, 2 × NCH_3_), 2.15–2.04 (m, 4H, 2 × CH_2_) 1.79 (s, 3H, CH_3_), 1.67 (s, 3H, CH_3_), 1.58 (s, 3H, CH_3_); ^13^C{^1^H} NMR (150 MHz, CDCl_3_): δ 155.1 (C, C-6), 152.5 (CH, C-2), 150.5 (C, C-4), 142.5 (C=), 137.9 (CH, C-8), 132.2 (C=), 123.7 (CH=), 120.3 (C, C-5), 117.9 (CH=), 41.1 (NCH_2_), 39.6 (CH_2_), 38.7 (2 × NCH_3_), 26.3 (CH_2_), 25.8 (CH_3_), 17.9 (CH_3_), 16.6 (CH_3_); IR (film from CH_2_Cl_2_): ν_max_ 2963, 2919, 1637 cm^−1^; HRESIMS *m*/*z*: [M + H]^+^ Calcd. for C_17_H_26_N_5_ 300.2183; Found 300.2184 (Δ = 0.3 ppm). HRESIMS/MS (40 eV) *m*/*z* (%): 164.0914 (100), 149.0683 (11), 121.0503 (16).

Compound **10**: 6.3 mg (12%), white powder; R*_f_* = 0.21 (5% MeOH/EA); ^1^H NMR (600 MHz, CDCl_3_): δ 8.00 (s, 1H, H-8), 7.95 (s, 1H, H-2), 5.48 (t, *J* = 6.9 Hz, 1H, CH=), 5.07–5.02 (m, 1H, CH=), 4.95 (d, *J* = 7.1 Hz, 2H, NCH_2_), 3.92 (br s, 3H, NCH_3_), 3.34 (br s, 3H, NCH_3_), 2.17–2.08 (m, 4H, 2 × CH_2_), 1.81 (s, 3H, CH_3_), 1.68 (s, 3H, CH_3_), 1.58 (s, 3H, CH_3_); ^13^C{^1^H} NMR (150 MHz, CDCl_3_): δ 153.4 (C, C-6), 152.6 (C, C-8), 150.7 (C, C-4), 144.4 (C=), 140.3 (CH, C-2), 132.4 (C=), 123.6 (CH=), 121.6 (C, C-5), 116.6 (CH=), 46.9 (NCH_2_), 39.9 (NCH_3_), 39.6 (CH_2_), 38.1 (NCH_3_), 26.3 (CH_2_), 25.8 (CH_3_), 17.9 (CH_3_), 16.7 (CH_3_); IR (film from CH_2_Cl_2_): ν_max_ 3077, 2964, 2922, 1607 cm^−1^; HRESIMS *m*/*z*: [M + H]^+^ Calcd. for C_17_H_26_N_5_ 300.2183; Found 300.2184 (Δ = 0.3 ppm); HRESIMS/MS (20 eV) *m*/*z* (%): 164.0914 (100).

#### 3.2.5. *N*,*N*-Dimethyl-9-((6*E*)-3,7,11-trimethyldodeca-2,6,10-trien-1-yl)-9*H*-purin-6-amine (**11**) and *N*,*N*-dimethyl-3-((6*E*)-3,7,11-trimethyldodeca-2,6,10-trien-1-yl)-3*H*-purin-6-amine (**12**)

6-(Dimethylamino)purine (0.088 mmol, 14.4 mg), K_2_CO_3_ (0.16 mmol, 22.3 mg) and farnesyl bromide (0.11 mmol, 31.6 mg) in DMF (1 mL) at 50 °C for 21 h yielded **11** and **12**.

Compound **11**: 5.6 mg (17%), white powder; R*_f_* = 0.29 (1:1 EA/PE); 3:2 *E*/*Z*, NMR data for major isomer: ^1^H NMR (500 MHz, CDCl_3_): δ 8.36 (s, 1H, H-2), 7.70 (s, 1H, H-8), 5.44 (t, *J* = 7.1 Hz, 1H, CH=), 5.11–5.03 (m, 2H, 2 × CH=), 4.75 (d, *J* = 7.1 Hz, 2H, NCH_2_), 3.53 (br s, 6H, 2 × NCH_3_), 2.16–2.06 (m, 4H, 2 × CH_2_), 2.06–1.99 (m, 2H, CH_2_), 1.99–1.92 (m, 2H, CH_2_), 1.81 (s, 3H, CH_3_), 1.68 (s, 3H, CH_3_), 1.58 (s, 6H, 2 × CH_3_); ^13^C{^1^H} NMR (150 MHz, CDCl_3_): δ 155.1 (C, C-6), 152.5 (CH, C-2), 150.5 (C, C-4), 142.5 (C=), 137.9 (CH, C-8), 135.9 (C=), 131.5 (C=), 124.4 (CH=), 120.3 (C, C-5), 117.9 (CH=), 41.2 (NCH_2_), 39.8 (CH_2_), 39.6 (CH_2_), 38.7 (2 × NCH_3_), 26.8 (CH_2_), 26.3 (CH_2_), 25.9 (CH_3_), 17.8 (CH_3_), 16.7 (CH_3_), 16.2 (CH_3_); IR (film from CH_2_Cl_2_): ν_max_ 3051, 2961, 2917, 2856, 1589 cm^−1^; HRESIMS *m*/*z*: [M + H]^+^ Calcd. for C_22_H_34_N_5_ 368.2809; Found 368.2817 (Δ = 2.2 ppm); HRESIMS/MS (20 eV) *m*/*z* (%): 164.0894 (100).

Compound **12**: 9.5 mg (29%), white powder; R*_f_* = 0.25 (10% MeOH/EA); 3:2 *E*/*Z*, NMR data for major isomer: ^1^H NMR (600 MHz, CDCl_3_): δ 8.01 (s, 1H, H-8), 7.95 (s, 1H, H-2), 5.48 (t, *J* = 7.3 Hz, 1H, CH=), 5.09–5.03 (m, 2H, 2 × CH=), 4.96 (d, *J* = 7.3 Hz, 2H, NCH_2_), 3.92 (br s, 3H, NCH_3_), 3.33 (br s, 3H, NCH_3_), 2.17–2.08 (m, 4H, 2 × CH_2_), 2.05–1.99 (m, 2H, CH_2_), 1.99–1.94 (m, 2H, CH_2_), 1.82 (s, 3H, CH_3_), 1.66 (s, 3H, CH_3_), 1.58 (s, 6H, 2 × CH_3_); ^13^C{^1^H} NMR (150 MHz, CDCl_3_): δ 153.4 (C, C-6), 152.4 (CH, C-8), 150.5 (C, C-4), 144.5 (C=), 140.4 (CH, C-2), 136.1 (C=), 131.5 (C=), 124.4 (CH=), 123.3 (CH=), 121.6 (C, C-5), 116.5 (CH=), 46.9 (NCH_2_), 39.9 (NCH_3_), 39.8 (CH_2_), 39.6 (CH_2_), 38.1 (NCH_3_), 26.8 (CH_2_), 26.2 (CH_2_), 25.8 (CH_3_), 17.8 (CH_3_), 16.8 (CH_3_), 16.2 (CH_3_); IR (film from CH_2_Cl_2_): ν_max_ 2963, 2924, 2856, 1608 cm^−1^; HRESIMS *m*/*z*: [M + H]^+^ Calcd. for C_22_H_34_N_5_ 368.2809; Found 368.2818 (Δ = 2.4 ppm); HRESIMS/MS (40 eV) *m*/*z* (%): 164.0932 (100), 81.0704 (14).

#### 3.2.6. *N*,*N*-Dimethyl-9-((2Z,6*E*,10*E*)-3,7,11,15-tetramethylhexadeca-2,6,10,14-tetraen-1-yl)-9*H*-purin-6-amine (**13**) and *N*,*N*-dimethyl-9-((2*E*,6*E*,10*E*)-3,7,11,15-tetramethylhexadeca-2,6,10,14-tetraen-1-yl)-9*H*-purin-6-amine (**14**)

6-(Dimethylamino)purine (0.14 mmol, 23.4 mg), K_2_CO_3_ (0.16 mmol, 22.7 mg) and geranylgeranyl bromide (0.15 mmol, 51.5 mg) in DMF (1 mL) for 44 h yielded **13** and **14**. 

Compound **13**: 4.5 mg (28%), white solid; R*_f_* = 0.21 (1:2 EA/PE); ^1^H NMR (500 MHz, CDCl_3_): δ 8.35 (s, 1H, H-2), 7.70 (s, 1H, H-8), 5.43 (t, *J* = 7.1 Hz, 1H, CH=), 5.15–5.04 (m, 3H, 3 × CH=), 4.75 (d, *J* = 7.1 Hz, 2H, NCH_2_), 3.53 (br s, 6H, 2 × NCH_3_), 2.27–2.20 (m, 2H, CH_2_), 2.18–2.10 (m, 2H, CH_2_), 2.10–2.01 (m, 4H, 2 × CH_2_), 2.01–1.91 (m, 4H, 2 × CH_2_), 1.79 (s, 3H, CH_3_), 1.67 (s, 3H, CH_3_), 1.61 (s, 3H, CH_3_), 1.59 (s, 6H, 2 × CH_3_); ^13^C{^1^H} NMR (150 MHz, CDCl_3_): δ 155.1 (C, C-6), 152.5 (CH, C-2), 150.5 (C, C-4), 142.4 (C=), 137.9 (CH, C-8), 136.3 (C=), 135.2 (C=), 131.4 (C=), 124.5 (CH=), 124.2 (CH=), 123.3 (CH=), 120.7 (C, C-5), 118.7 (CH=), 41.0 (NCH_2_), 39.9 (CH_2_), 39.8 (CH_2_), 38.6 (2 × NCH_3_), 32.3 (CH_2_), 26.9 (CH_2_), 26.7 (CH_2_), 26.5 (CH_2_), 25.9 (CH_3_), 23.6 (CH_3_), 17.8 (CH_3_), 16.20 (CH_3_), 16.16 (CH_3_); IR (film from CH_2_Cl_2_): ν_max_ 3043, 2921, 2854, 1590 cm^−1^; HRESIMS *m*/*z*: [M + H]^+^ Calcd. for C_27_H_42_N_5_ 436.3435; Found 436.3430 (Δ = −1.1 ppm); HRESIMS/MS (40 eV) *m*/*z* (%): 164.0912 (100).

Compound **14**: 6.7 mg (14%), white solid; R*_f_* = 0.18 (1:2 EA/PE); ^1^H NMR (500 MHz, CDCl_3_): δ 8.36 (s, 1H, H-2), 7.70 (s, 1H, H-8), 5.43 (t, *J* = 7.0 Hz, 1H, CH=), 5.12–5.04 (m, 3H, 3 × CH=), 4.75 (d, *J* = 7.1 Hz, 2H, NCH_2_), 3.56 (br s, 6H, 2 × NCH_3_), 2.15–2.08 (m, 4H, 2 × CH_2_), 2.08–2.01 (m, 4H, 2 × CH_2_), 1.99–1.93 (m, 4H, 2 × CH_2_), 1.80 (s, 3H, CH_3_), 1.67 (s, 3H, CH_3_), 1.59 (s, 3H, CH_3_), 1.58 (s, 6H, 2 × CH_3_); ^13^C{^1^H} NMR (150 MHz, CDCl_3_): δ 155.1 (C, C-6), 152.5 (CH, C-2), 150.5 (C, C-4), 142.5 (C=), 137.9 (CH, C-8), 135.9 (C=), 135.2 (C=), 131.4 (C=), 124.5 (CH=), 124.2 (CH=), 123.5 (CH=), 120.3 (C, C-5), 117.9 (CH=), 41.1 (NCH_2_), 39.9 (CH_2_), 39.8 (CH_2_), 39.6 (CH_2_), 38.6 (2 × NCH_3_), 26.9 (CH_2_), 26.7 (CH_2_), 26.3 (CH_2_), 25.8 (CH_3_), 17.8 (CH_3_), 16.7 (CH_3_), 16.2 (CH_3_), 16.1 (CH_3_); IR (film from CH_2_Cl_2_): ν_max_ 3104, 2962, 2917, 2855, 1590 cm^−1^; HRESIMS *m*/*z*: [M + H]^+^ Calcd. for C_27_H_42_N_5_ 436.3435; Found 436.3466 (Δ = 7.1 ppm); HRESIMS/MS (40 eV) *m*/*z* (%): 164.0873 (100).

#### 3.2.7. (*E*)-6-Chloro-9-(3,7-dimethylocta-2,6-dien-1-yl)-9*H*-purine (**15**) and (*E*)-6-chloro-7-(3,7-dimethylocta-2,6-dien-1-yl)-7*H*-purine (**16**)

Following the general alkylation procedure, also previously published using 80 °C [54], 6-chloropurine (0.30 mmol, 45.9 mg), K_2_CO_3_ (0.62 mmol, 85.6 mg) and geranyl bromide (0.33 mmol, 71.8 mg) in DMF (1 mL) for 21 h yielded **15** and **16**.

Compound **15**: 39.4 mg (46%), colourless oil; R*_f_* = 0.48 (2:3 EA/PE); IR data and select ^1^H NMR data previously reported in CD_3_OD [54]; ^1^H NMR (300 MHz, CDCl_3_): δ 8.72 (s, 1H, H-2), 8.07 (s, 1H, H-8), 5.42 (t, *J* = 7.2 Hz, 1H, CH=), 5.05–4.97 (m, 1H, CH=), 4.84 (d, *J* = 7.3 Hz, 2H, NCH_2_), 2.09 (s, 4H, 2 × CH_2_), 1.81 (s, 3H, CH_3_), 1.64 (s, 3H, CH_3_), 1.55 (s, 3H, CH_3_); ^13^C{^1^H} NMR (150 MHz, CDCl_3_): δ 151.9 (CH, C-2), 151.8 (C, C-6), 150.9 (C, C-4), 144.8 (CH, C-8), 144.1 (C=), 132.4 (C, C-5), 131.8 (C=), 123.4 (CH=), 116.7 (CH=), 41.9 (NCH_2_), 39.5 (CH_2_), 26.1 (CH_2_), 25.8 (CH_3_), 17.8 (CH_3_), 16.7 (CH_3_); HRESIMS *m*/*z*: [M + H]^+^ Calcd. for C_15_H_20_ClN_4_ 291.1371; Found 291.1371 (Δ = 0.0 ppm); HRESIMS/MS (40 eV) *m*/*z* (%): 157.0074 (22), 155.0105 (57), 119.0346 (100).

Compound **16**: 17.2 mg (20%), colourless oil; R*_f_* = 0.16 (2:3 EA/PE); IR data and select ^1^H NMR data previously reported in CD_3_OD [54]; ^1^H NMR (300 MHz, CDCl_3_): δ 8.85 (s, 1H, H-2), 8.23 (s, 1H, H-8), 5.43 (t, *J* = 6.8 Hz, 1H, CH=), 5.07 (d, *J* = 7.0 Hz, 2H, NCH_2_), 5.05–5.00 (m, 1H, CH=), 2.13 (s, 4H, 2 × CH_2_), 1.80 (s, 3H, CH_3_), 1.66 (s, 3H, CH_3_), 1.58 (s, 3H, CH_3_); ^13^C{^1^H} NMR (150 MHz, CDCl_3_): δ 162.2 (C, C-6), 152.5 (CH, C-2), 148.4 (CH, C-8), 144.2 (C=), 143.3 (C, C-4), 132.6 (C=), 123.3 (CH=), 122.7 (C, C-5), 117.1 (CH=), 45.4 (NCH_2_), 39.5 (CH_2_), 26.1 (CH_2_), 25.8 (CH_3_), 17.9 (CH_3_), 16.8 (CH_3_); HRESIMS *m*/*z*: [M + H]^+^ Calcd. for C_15_H_20_ClN_4_ 291.1371; Found 291.1365 (Δ = −2.1 ppm); HRESIMS/MS (40 eV) *m*/*z* (%): 157.0062 (100), 155.0081 (40).

#### 3.2.8. 6-Chloro-9-((6*E*)-3,7,11-trimethyldodeca-2,6,10-trien-1-yl)-9*H*-purine (**17**) and 6-chloro-7-((2*E*,6*E*)-3,7,11-trimethyldodeca-2,6,10-trien-1-yl)-9*H*-purine (**18**)

6-Chloropurine (0.31 mmol, 48.5 mg), K_2_CO_3_ (0.6 mmol, 83 mg) and farnesyl bromide (0.33 mmol, 94.0 mg) in DMF (2 mL) at 50 °C for 24 h yielded **17** and **18**.

Compound **17**: 47.4 mg (43%), colourless oil; R*_f_* = 0.24 (1:2 EA/PE); 2:1 *E*/*Z*, NMR data for major isomer: ^1^H NMR (600 MHz, CDCl_3_): δ 8.72 (s, 1H, H-2), 8.08 (s, 1H, H-8), 5.42 (t, *J* = 7.2 Hz, 1H, CH=), 5.06–5.00 (m, 2H, 2 × CH=), 4.85 (d, *J* = 7.3 Hz, 2H, NCH_2_), 2.13–2.07 (m, 4H, 2 × CH_2_), 2.02–1.96 (m, 2H, CH_2_), 1.95–1.90 (m, 2H, CH_2_), 1.82 (s, 3H, CH_3_), 1.63 (s, 3H, CH_3_), 1.55 (s, 6H, 2 × CH_3_); ^13^C{^1^H} NMR (150 MHz, CDCl_3_): δ 152.0 (C, C-2), 151.8 (C, C-6), 151.0 (C, C-4), 144.9 (C, C-8), 144.1 (C=), 136.0 (C=), 132.3 (C, C-5), 131.5 (C=), 124.3 (CH=), 123.3 (CH=), 116.6 (CH=), 41.9 (NCH_2_), 39.8 (CH_2_), 39.5 (CH_2_), 26.8 (CH_2_), 26.2 (CH_2_), 25.8 (CH_3_), 17.8 (CH_3_), 16.8 (CH_3_), 16.2 (CH_3_); IR (film from CH_2_Cl_2_): ν_max_ 3115, 2969, 2930, 1335, 939, 637 cm^−1^; HRESIMS *m*/*z*: [M + H]^+^ Calcd. for C_20_H_28_ClN_4_ 359.1997; Found 359.1993 (Δ = −1.1 ppm); HRESIMS/MS (40 eV) *m*/*z* (%): 157.0073 (40), 155.0103 (100), 119.0343 (47), 81.0697 (47). 

Compound **18**: 21.3 mg (19%), colourless oil; R*_f_* = 0.20 (1:1 EA/PE); *E*/*Z* 3:2, NMR data for major isomer: ^1^H NMR (600 MHz, CDCl_3_): δ 8.86 (s, 1H, H-2), 8.23 (s, 1H, H-8), 5.44 (t, *J* = 7.0 Hz, 1H, CH=), 5.08 (d, *J* = 7.1 Hz, 2H, NCH_2_), 5.07–5.04 (m, 2H, 2 × CH=), 2.18–2.09 (m, 4H, 2 × CH_2_), 2.05–1.99 (m, 2H, CH_2_), 1.98–1.93 (m, 2H, CH_2_), 1.82 (s, 3H, CH_3_), 1.65 (s, 3H, CH_3_), 1.58 (s, 6H, 2 × CH_3_); ^13^C{^1^H} NMR (150 MHz, CDCl_3_): δ 162.0 (C, C-6), 152.4 (CH, C-2), 148.5 (CH, C-8), 144.2 (C=), 143.3 (C, C-4), 136.2 (C=), 131.5 (C=), 124.2 (CH=), 123.2 (CH=), 122.6 (C, C-5), 117.0 (CH=), 45.4 (NCH_2_), 39.8 (CH_2_), 39.5 (CH_2_), 26.8 (CH_2_), 26.2 (CH_2_), 25.8 (CH_3_), 17.8 (CH_3_), 16.9 (CH_3_), 16.2 (CH_3_); IR (film from CH_2_Cl_2_): ν_max_ 3053, 2981, 1264, 732 cm^−1^; HRESIMS *m*/*z*: [M + H]^+^ Calcd. for C_20_H_28_ClN_4_ 359.1997; Found 359.2002 (Δ = 1.4 ppm); HRESIMS/MS (40 eV) *m*/*z* (%): 157.0001 (27), 155.0031 (66), 119.0287 (31), 95.0806 (23), 93.0651 (13), 81.0660 (100), 79.0539 (15). 

#### 3.2.9. 6-Chloro-9-((2*E*,6*E*,10*E*)-3,7,11,15-tetramethylhexadeca-2,6,10,14-tetraen-1-yl)-9*H*-purine (**19**), 6-chloro-7-((2*E*,6*E*,10*E*)-3,7,11,15-tetramethylhexadeca-2,6,10,14-tetraen-1-yl)-7*H*-purine (**20**), and 6-chloro-7-((2*Z*,6*E*,10*E*)-3,7,11,15-tetramethylhexadeca-2,6,10,14-tetraen-1-yl)-7*H*-purine (**21**)

Following the general alkylation procedure, also previously published [55], 6-chloropurine (0.30 mmol, 45.6 mg), K_2_CO_3_ (0.35 mmol, 48.2 mg) and geranylgeranyl bromide (0.32 mmol, 111 mg) in DMF (1 mL) for 27 h yielded **19**, **20**, and **21_._**

Compound **19**: 19.4 mg (19%), colourless oil; R*_f_* = 0.51 (2:3 EA/PE); ^1^H and ^13^C NMR data previously reported [55]; IR (film from CH_2_Cl_2_): ν_max_ 3070, 2966, 2922, 2855, 1592, 1560, 1335 cm^−1^; HRESIMS *m*/*z*: [M + H]^+^ Calcd. for C_25_H_36_ClN_4_ 427.2623; Found 427.2614 (Δ = −2.1 ppm); HRESIMS/MS (40 eV) *m*/*z* (%): 157.0068 (39), 155.0098 (100), 119.0334 (21).

Compound **20**: 15.2 mg (15%), colourless oil; R*_f_* = 0.21 (2:3 EA/PE); ^1^H and ^13^C NMR data previously reported [55]; IR (film from CH_2_Cl_2_): ν_max_ 3053, 2971, 2931, 733 cm^−1^; HRESIMS *m*/*z*: [M + H]^+^ Calcd. for C_25_H_36_ClN_4_ 427.2623; Found 427.2642 (Δ = 4.4 ppm).

Compound **21**: 16.8 mg (49%), colourless oil; R*_f_* = 0.3 (2:3 EA/PE); ^1^H NMR (500 MHz, CDCl_3_): δ 8.87 (s, 1H, H-2), 8.23 (s, 1H, H-8), 5.45 (t, *J* = 7.1 Hz, 1H, CH=), 5.15–5.08 (m, 3H, 3 × CH=), 5.07 (d, *J* = 7.1 Hz, 2H, NCH_2_), 2.28–2.21 (m, 2H, CH_2_), 2.21–2.13 (m, 2H, CH_2_), 2.10–2.02 (m, 4H, 2 × CH_2_), 2.02–1.93 (m, 4H, 2 × CH_2_), 1.84 (s, 3H, CH_3_), 1.67 (s, 3H, CH_3_), 1.62 (s, 3H, CH_3_), 1.59 (s, 6H, 2 × CH_3_); ^13^C{^1^H} NMR (150 MHz, CDCl_3_): δ 162.1 (C, C-6), 152.5 (CH, C-2), 148.4 (CH, C-8), 144.1 (C=), 143.2 (C, C-4), 136.9 (C=), 135.4 (C=), 131.3 (C=), 124.4 (CH=), 124.1 (CH=), 122.9 (CH=), 122.5 (C, C-5), 117.9 (CH=), 45.2 (NCH_2_), 39.9 (CH_2_), 39.8 (CH_2_), 32.5 (CH_2_), 26.9 (CH_2_), 26.7 (CH_2_), 26.3 (CH_2_), 25.8 (CH_3_), 23.6 (CH_3_), 17.8 (CH_3_), 16.18 (CH_3_), 16.16 (CH_3_); IR (film from CH_2_Cl_2_): ν_max_ 3055, 2973, 2932, 733 cm^−1^_;_ HRESIMS *m*/*z*: [M + H]^+^ Calcd. for C_25_H_36_ClN_4_ 427.2623; Found 427.2634 (Δ = 2.6 ppm); HRESIMS/MS (40 eV) *m*/*z* (%): 157.0086 (32), 155.0118 (100), 121.10185 (10), 119.03516 (16), 109.1013 (11), 107.0858 (15).

#### 3.2.10. 3,7-Bis((*E*)-3,7-dimethylocta-2,6-dien-1-yl)-3,7-dihydro-1*H*-purine-2,6-dione (**27**)

Xanthine (2.1 mmol, 311.5 mg), K_2_CO_3_ (2.0 mmol, 272.2 mg) and geranyl bromide (2.4 mmol, 521 mg) in DMF (3 mL) for 48 h yielded **27**, with modified work up—concentrating under reduced pressure, 44.6 mg (9%), colourless oil. R*_f_* = 0.15 (1:2 EA/PE); ^1^H NMR (600 MHz, CDCl_3_): δ 8.10 (br s, 1H, NH), 7.55 (s, *J* = 2.8 Hz, 1H, H-8), 5.43 (t, *J* = 6.7 Hz, 1H, CH=), 5.34 (t, *J* = 6.4 Hz, 1H, CH=), 5.07–5.01 (m, 2H, 2 × CH=), 4.89 (d, *J* = 7.3 Hz, 2H, NCH_2_), 4.66 (d, *J* = 6.9 Hz, 2H, NCH_2_), 2.15–2.09 (m, 4H, 2 × CH_2_), 2.09–2.03 (m, 2H, CH_2_), 2.02–1.96 (m, 2H, CH_2_), 1.84 (s, 3H, CH_3_), 1.77 (s, 3H, CH_3_), 1.68 (s, 3H, CH_3_), 1.63 (s, 3H, CH_3_), 1.59 (s, 3H, CH_3_), 1.56 (s, 3H, CH_3_); ^13^C{^1^H} NMR (150 MHz, CDCl_3_): δ 154.7 (C, C-6), 150.62 (C, C-2 or C-4), 150.58 (C, C-2 or C-4), 143.8 (C=), 140.8 (CH=), 140.7 (CH, C-8), 132.4 (C=), 131.8 (C=), 124.0 (CH=), 123.6 (CH=), 118.0 (CH=), 117.2 (CH=), 107.5 (C, C-5), 44.7 (NCH_2_), 40.9 (NCH_2_), 39.7 (CH_2_), 39.6 (CH_2_), 26.5 (CH_2_), 26.3 (CH_2_), 25.84 (CH_3_), 25.80 (CH_3_), 17.9 (CH_3_), 17.8 (CH_3_), 16.7 (CH_3_), 16.6 (CH_3_); IR (film from CH_2_Cl_2_): ν_max_ 3423, 3176, 3115, 3052, 2967, 2926, 1679 cm^−1^; HRESIMS *m*/*z*: [M + H]^+^ Calcd. for C_25_H_37_N_4_O_2_ 425.2911; Found 425.2910 (Δ = −0.2 ppm); HRESIMS/MS (10 eV) *m*/*z* (%): 289.1638 (70), 153.0395 (100).

#### 3.2.11. 3,7-Bis((6*E*)-3,7,11-trimethyldodeca-2,6,10-trien-1-yl)-3,7-dihydro-1*H*-purine-2,6-dione (**28**)

Xanthine (2.0 mmol, 306.3 mg), K_2_CO_3_ (3.0 mmol, 420.3 mg) and farnesyl bromide (2.4 mmol, 683.4 mg) in DMF (3 mL) for 25 h yielded **28**, 19.3 mg (3%), colourless oil. R*_f_* = 0.25 (1:2 EA/PE); 3:2 *E*/*Z*, NMR data for major isomer: ^1^H NMR (600 MHz, CDCl_3_): δ 8.57 (br s, 1H, NH), 7.54 (s, 1H, H-8), 5.43 (t, *J* = 6.6 Hz, 1H, CH=), 5.34 (t, *J* = 6.8 Hz, 1H, CH=), 5.12–5.02 (m, 4H, 4 × CH=), 4.89 (d, *J* = 7.2 Hz, 2H, NCH_2_), 4.66 (d, *J* = 6.9 Hz, 2H, NCH_2_), 2.16–1.89 (m, 16H, 8 × CH_2_), 1.85 (s, 3H, CH_3_), 1.78 (s, 3H, CH_3_), 1.66 (s, 6H, 2 × CH_3_), 1.58 (s, 9H, 3 × CH_3_), 1.55 (s, 3H, CH_3_); ^13^C{^1^H} NMR (150 MHz, CDCl_3_): δ 154.9 (C, C-6), 150.7 (C, C-2 or C-4), 150.6 (C, C-2 or C-4), 143.8 (C=), 141.1 (C=), 140.8 (CH, C-8), 136.0 (C=), 135.4 (C=), 131.5 (C=), 131.4 (C=), 124.4 (CH=), 124.3 (CH=), 123.9 (CH=), 123.4 (CH=), 118.0 (CH=), 117.2 (CH=), 107.5 (C, C-5), 44.7 (NCH_2_), 40.8 (NCH_2_), 39.79 (CH_2_), 39.78 (CH_2_), 39.7 (CH_2_), 39.6 (CH_2_), 26.81 (CH_2_), 26.78 (CH_2_), 26.4 (CH_2_), 26.3 (CH_2_), 25.84 (CH_3_), 25.83 (CH_3_), 17.82 (CH_3_), 17.81 (CH_3_), 16.74 (CH_3_), 16.67 (CH_3_), 16.2 (CH_3_), 16.1 (CH_3_); IR (film from CH_2_Cl_2_): ν_max_ 3166, 3065, 2964, 2927, 2856, 1686 cm^−1^; HRESIMS *m*/*z*: [M + H]^+^ Calcd. for C_35_H_53_N_4_O_2_ 561.4163; Found 561.4169 (Δ = 1.1 ppm); HRESIMS/MS (20 eV) *m*/*z* (%): 561.1128 (39), 357.2262 (48), 153.0391 (100).

#### 3.2.12. (*E*)-7-(3,7-Dimethylocta-2,6-dien-1-yl)-3-methyl-3,7-dihydro-1*H*-purine-2,6-dione (**29**)

3-Methylxanthine (0.17 mmol, 28.0 mg), K_2_CO_3_/Na_2_CO_3_ (1:1, 48.6 mg) and geranyl bromide (0.36 mmol, 77 mg) in DMF (2 mL) at 50 °C for 48 h yielded **29**, with modified work up—the concentrated reaction was filtered from DCM, and additionally recrystallised from PE after chromatography, 16.9 mg (5%), white powder. R*_f_* = 0.2 (1:2 EA/PE); ^1^H NMR (600 MHz, CDCl_3_): δ 8.10 (s, 1H, NH), 7.55 (s, 1H, H-8), 5.43 (t, *J* = 7.3 Hz, 1H, CH=), 5.07–5.03 (m, 1H, CH=), 4.90 (d, *J* = 7.3 Hz, 2H, NCH_2_), 3.55 (s, 3H, NCH_3_), 2.16–2.08 (m, 4H, 2 × CH_2_), 1.78 (s, 3H, CH_3_), 1.68 (s, 3H, CH_3_), 1.59 (s, 3H, CH_3_); ^13^C{^1^H} NMR (150 MHz, CDCl_3_): δ 154.5 (C, C-6), 150.99 (C, C-2), 150.96 (C, C-4), 143.9 (C=), 140.8 (CH, C-8), 132.4 (C=), 123.5 (CH=), 117.1 (CH=), 107.4 (C, C-5), 44.8 (NCH_2_), 39.6 (CH_2_), 29.2 (NCH_3_), 26.2 (CH_2_), 25.9 (CH_3_), 17.9 (CH_3_), 16.6 (CH_3_); IR (film from CH_2_Cl_2_): ν_max_ 3121, 3021, 2964, 2916, 2826, 1678 cm^−1^; HRESIMS *m*/*z*: [M + H]^+^ Calcd. for C_16_H_23_N_4_O_2_ 303.1816; Found 303.1812 (Δ = −1.3 ppm); HRESIMS/MS (40 eV) *m*/*z* (%): 167.0538 (100), 149.0431 (8), 124.0487 (17).

#### 3.2.13. 3-Methyl-7-((2*Z*,6*E*)-3,7,11-trimethyldodeca-2,6,10-trien-1-yl)-3,7-dihydro-1*H*-purine-2,6-dione (**30**) and 3-methyl-7-((2*E*,6*E*)-3,7,11-trimethyldodeca-2,6,10-trien-1-yl)-3,7-dihydro-1*H*-purine-2,6-dione (**31**)

3-Methylxanthine (0.20 mmol, 32.9 mg), K_2_CO_3_ (0.37 mmol, 51.2 mg) and farnesyl bromide (0.34 mmol, 98 mg) in DMF (2 mL) for 21 h yielded **30** and **31**. 

Compound **30**: 4.6 mg (8%), white solid; R*_f_* = 0.20 (1:1 EA/PE); ^1^H NMR (600 MHz, CDCl_3_): δ 8.10 (s, 1H, NH), 7.55 (s, 1H, H-8), 5.44 (t, *J* = 6.9 Hz, 1H, CH=), 5.11–5.05 (m, 2H, 2 × CH=), 4.88 (d, *J* = 7.4 Hz, 2H, NCH_2_), 3.54 (s, 3H, NCH_3_), 2.23–2.17 (m, 2H, CH_2_), 2.15–2.10 (m, 2H, CH_2_), 2.07–2.01 (m, 2H, CH_2_), 1.99–1.93 (m, 2H, CH_2_), 1.81 (s, 3H, CH_3_), 1.67 (s, 3H, CH_3_), 1.60 (s, 3H, CH_3_), 1.59 (s, 3H, CH_3_); ^13^C{^1^H} NMR (150 MHz, CDCl_3_): δ 154.5 (C, C-6), 151.0 (C, C-2), 150.9 (C, C-4), 143.7 (C=), 140.8 (CH, C-8), 136.5 (C=), 131.6 (C=), 124.3 (CH=), 123.1 (CH=), 118.0 (CH=), 107.4 (C, C-5), 44.6 (NCH_2_), 39.8 (CH_2_), 32.2 (CH_2_), 29.2 (NCH_3_), 26.7 (CH_2_), 26.4 (CH_2_), 25.9 (CH_3_), 23.6 (CH_3_), 17.8 (CH_3_), 16.2 (CH_3_); IR (film from CH_2_Cl_2_): ν_max_ 3400, 3162, 3035, 2969, 2930, 1683 cm^−1^; HRESIMS *m*/*z*: [M + H]^+^ Calcd. for C_21_H_31_N_4_O_2_ 371.2442; Found 371.2452 (Δ = 2.7 ppm).

Compound **31**: 33.1 mg (45%), white solid; R*_f_* = 0.18 (1:1, EA/PE); 2:1 *E*/*Z*, NMR data for major isomer: ^1^H NMR (600 MHz, CDCl_3_): δ 8.89 (s, 1H, NH), 7.56 (s, 1H, H-8), 5.42 (t, *J* = 7.3 Hz, 1H, CH=), 5.08–5.04 (m, 2H, 2 × CH=), 4.90 (d, *J* = 7.3 Hz, 2H, NCH_2_), 3.54 (s, 3H, NCH_3_), 2.16–2.07 (m, 4H, 2 × CH_2_), 2.06–1.99 (m, 2H, CH_2_), 1.98–1.93 (m, 2H, CH_2_), 1.78 (s, 3H, CH_3_), 1.66 (s, 3H, CH_3_), 1.58 (s, 6H, 2 × CH_3_); ^13^C{^1^H} NMR (150 MHz, CDCl_3_): δ 154.9 (C, C-6), 151.3 (C, C-2), 150.9 (C, C-4), 143.9 (C=), 140.8 (CH, C-8), 136.0 (C=), 131.5 (C=), 124.3 (CH=), 123.4 (CH=), 117.1 (CH=), 107.5 (C, C-5), 44.7 (NCH_2_), 39.8 (CH_2_), 39.6 (CH_2_), 29.2 (NCH_3_), 26.8 (CH_2_), 26.2 (CH_2_), 25.8 (CH_3_), 17.8 (CH_3_), 16.7 (CH_3_), 16.2 (CH_3_); HRESIMS *m*/*z*: [M + H]^+^ Calcd. for C_21_H_31_N_4_O_2_ 371.2442; Found 371.2445 (Δ = 0.8 ppm); HRESIMS/MS (40 eV) *m*/*z* (%): 167.0552 (100), 124.0506 (87).

#### 3.2.14. 3-Methyl-7-((2*Z*,6*E*,10*E*)-3,7,11,15-tetramethylhexadeca-2,6,10,14-tetraen-1-yl)-3,7-dihydro-1*H*-purine-2,6-dione (**32**) and 3-methyl-7-((2*E*,6*E*,10*E*)-3,7,11,15-tetramethylhexadeca-2,6,10,14-tetraen-1-yl)-3,7-dihydro-1*H*-purine-2,6-dione (**33**)

3-Methylxanthine (0.35 mmol, 40.9 mg), K_2_CO_3_ (0.38 mmol, 52.5 mg) and geranylgeranyl bromide (0.37 mmol, 130 mg) in DMF (1 mL) for 44 h yielded **32** and **33**. 

Compound **32**: 10.5 mg (26%), white, waxy solid; R*_f_* = 0.20 (1:1, EA/PE); ^1^H NMR (600 MHz, CDCl_3_): δ 8.45 (s, 1H, NH), 7.55 (s, 1H, H-8), 5.44 (t, *J* = 7.1 Hz, 1H, CH=), 5.12–5.05 (m, 3H, 3 × CH=), 4.88 (d, *J* = 7.2 Hz, 2H, NCH_2_), 3.54 (s, 3H, NCH_3_), 2.23–2.18 (m, 2H, CH_2_), 2.15–2.09 (m, 2H, CH_2_), 2.08–2.01 (m, 4H, 2 × CH_2_), 2.00–1.93 (m, 4H, 2 × CH_2_), 1.80 (s, 3H, CH_3_), 1.67 (s, 3H, CH_3_), 1.60 (s, 3H, CH_3_), 1.59 (s, 6H, 2 × CH_3_); ^13^C{^1^H} NMR (150 MHz, CDCl_3_): δ 154.6 (C, C-6), 151.1 (C, C-2), 150.9 (C, C-4), 143.7 (C=), 140.8 (CH, C-8), 136.5 (C=), 135.3 (C=), 131.4 (C=), 124.5 (CH=), 124.1 (CH=), 123.1 (CH=), 118.0 (CH=), 107.4 (C, C-5), 44.6 (NCH_2_), 39.9 (CH_2_), 39.8 (CH_2_), 32.3 (CH_2_), 29.2 (NCH_3_), 26.9 (CH_2_), 26.7 (CH_2_), 26.5 (CH_2_), 25.8 (CH_3_), 23.6 (CH_3_), 17.8 (CH_3_), 16.20 (CH_3_), 16.15 (CH_3_); IR (film from CH_2_Cl_2_): ν_max_ 3458, 3159, 2968, 2924, 2852 cm^−1^; HRESIMS *m*/*z*: [M + H]^+^ Calcd. for C_26_H_39_N_4_O_2_ 439.3068; Found 439.3036 (Δ = −7.2 ppm); HRESIMS/MS (40 eV) *m*/*z* (%): 168.0565 (9), 167.0533 (100), 124.0494 (43).

Compound **33**: 22.1 mg (18%), white waxy solid; R*_f_* = 0.12 (1:1 EA/PE); ^1^H NMR (500 MHz, CDCl_3_): δ 8.94 (s, 1H, NH), 7.56 (s, 1H, H-8), 5.43 (t, *J* = 7.3 Hz, 1H, CH=), 5.12–5.04 (m, 3H, 3 × CH=), 4.90 (d, *J* = 7.3 Hz, 2H, NCH_2_), 3.54 (s, 3H, NCH_3_), 2.16–2.08 (m, 4H, 2 × CH_2_), 2.07–2.00 (m, 4H, 2 × CH_2_), 2.00–1.93 (m, 4H, 2 × CH_2_), 1.78 (s, 3H, CH_3_), 1.66 (s, 3H, CH_3_), 1.59 (s, 9H, 3 × CH_3_); ^13^C{^1^H} NMR (150 MHz, CDCl_3_): δ 154.9 (C, C-6), 151.3 (C-2), 150.9 (C-4), 143.9 (C=), 140.8 (CH, C-8), 136.0 (C=), 135.2 (C=), 131.41 (C=), 124.5 (CH=), 124.2 (CH=), 123.4 (CH=), 117.1 (CH=), 107.5 (C, C-5), 44.8 (NCH_2_), 39.84 (CH_2_), 39.79 (CH_2_), 39.6 (CH_2_), 29.2 (NCH_3_), 26.9 (CH_2_), 26.7 (CH_2_), 26.3 (CH_2_), 25.8 (CH_3_), 17.8 (CH_3_), 16.7 (CH_3_), 16.2 (CH_3_), 16.1 (CH_3_); IR (film from CH_2_Cl_2_): ν_max_ 3158, 3121, 3029, 2965, 2924, 2834, 1713, 1679 cm^−1^; HRESIMS *m*/*z*: [M + H]^+^ Calcd. for C_26_H_39_N_4_O_2_ 439.3068; Found 439.3035 (Δ = −7.5 ppm); HRESIMS/MS (40 eV) *m*/*z* (%): 168.0459 (9), 167.0437 (100), 124.0412 (39).

#### 3.2.15. (*E*)-7-(3,7-Dimethylocta-2,6-dien-1-yl)-1,3-dimethyl-3,7-dihydro-1*H*-purine-2,6-dione (**34**)

Theophylline hydrate (2.0 mmol, 391.3 mg), K_2_CO_3_/Na_2_CO_3_ (1:1, 244 mg) and geranyl bromide (4.0 mmol, 869 mg) in DMF (3 mL) for 3 h yielded **34**, with modified work up and modified purification—H_2_O (9 mL) was added to the reaction and the resulting precipitate was isolated and recrystallised from PE, 277.5 mg (45%), white crystals. ^1^H NMR (600 MHz, CDCl_3_): δ 7.53 (s, 1H, H-8), 5.43 (t, *J* = 7.6 Hz, 1H, CH=), 5.05 (t, *J* = 6.2 Hz, 1H, CH=), 4.93 (d, *J* = 7.3 Hz, 2H, NCH_2_), 3.59 (s, 3H, N(3)CH_3_), 3.42 (s, 3H, N(1)CH_3_), 2.15–2.07 (m, 4H, 2 × CH_2_), 1.78 (s, 3H, CH_3_), 1.68 (s, 3H, CH_3_), 1.59 (s, 3H, CH_3_); ^13^C{^1^H} NMR (150 MHz, CDCl_3_): δ 155.5 (C, C-6), 151.9 (C, C-2), 149.0 (C, C-4), 143.5 (C=), 140.3 (CH, C-8), 132.4 (C=), 123.6 (CH=), 117.5 (CH=), 107.2 (C, C-5), 44.7 (NCH_2_), 39.6 (CH_2_), 29.9 (N(3)CH_3_), 28.1 (N(1)CH_3_), 26.3 (CH_2_), 25.9 (CH_3_), 17.9 (CH_3_), 16.6 (CH_3_); IR (neat): ν_max_ 3098, 2964, 2926, 2855, 1695, 1645 cm^−1^; HRESIMS *m*/*z*: [M + H]^+^ Calcd. for C_17_H_25_N_4_O_2_ 317.1972; Found 317.1978 (Δ = 1.9 ppm); HRESIMS/MS (40 eV) *m*/*z* (%): 181.0714 (52), 124.0511 (100).

#### 3.2.16. 1,3-Dimethyl-7-((2*E*,6*E*)-3,7,11-trimethyldodeca-2,6,10-trien-1-yl)-3,7-dihydro-1*H*-purine-2,6-dione (**35**)

Theophylline hydrate (1.85 mmol, 370 mg), K_2_CO_3_/Na_2_CO_3_ (1:1, 170 mg) and farnesyl bromide (2.0 mmol, 570.5 mg). in DMF (2 mL) at 80 °C for 5 h yielded **35**, after modified work up and modified purification as per **34**, 333.1 mg (72%), white crystals. ^1^H NMR (600 MHz, CDCl_3_): δ 7.53 (s, 1H, H-8), 5.43 (t, *J* = 7.3 Hz, 1H, CH=), 5.09–5.05 (m, 2H, 2 × CH=), 4.93 (d, *J* = 7.2 Hz, 2H, NCH_2_), 3.58 (s, 3H, N(3)CH_3_), 3.41 (s, 3H, N(1)CH_3_), 2.16–2.07 (m, 4H, 2 × CH_2_), 2.03 (m, 2H, CH_2_), 1.99–1.93 (m, 2H, CH_2_), 1.79 (s, 3H, CH_3_), 1.67 (s, 1H, CH_3_), 1.60 (s, 3H, CH_3_), 1.59 (s, 3H, CH_3_); ^13^C{^1^H} NMR (150 MHz, CDCl_3_): δ 155.5 (C, C-6), 151.9 (C, C-2), 149.0 (C, C-4), 143.5 (C=), 140.3 (CH, C-8), 136.0 (C=), 131.6 (C=), 124.3 (CH=), 123.4 (CH=), 117.5 (CH=), 107.2 (C, C-5), 44.7 (NCH_2_), 39.8 (CH_2_), 39.6 (CH_2_), 29.9 (N(3)CH_3_), 28.1 (N(1)CH_3_), 26.8 (CH_2_), 26.3 (CH_2_), 25.9 (CH_3_), 17.8 (CH_3_), 16.7 (CH_3_), 16.2 (CH_3_); IR (film from CH_2_Cl_2_): ν_max_ 3097, 2695, 2922, 1695, 1646 cm^−1^; HRESIMS *m*/*z*: [M + H]^+^ Calcd. for C_22_H_33_N_4_O_2_ 385.2598; Found 385.2604 (Δ = 1.6 ppm); HRESIMS/MS (40 eV) *m*/*z* (%):181.0718 (100), 124.0506 (73).

#### 3.2.17. 1,3-Dimethyl-7-((2*Z*,6*E*,10*E*)-3,7,11,15-tetramethylhexadeca-2,6,10,14-tetraen-1-yl)-3,7-dihydro-1*H*-purine-2,6-dione (**36**) and 1,3-dimethyl-7-((2*E*,6*E*,10*E*)-3,7,11,15-tetramethylhexadeca-2,6,10,14-tetraen-1-yl)-3,7-dihydro-1*H*-purine-2,6-dione (**37**)

Theophylline hydrate (0.30 mmol, 59.1 mg), K_2_CO_3_ (0.34 mmol, 46.5 mg) and geranylgeranyl bromide (0.32 mmol, 111 mg) in DMF (1 mL) for 25 h yielded **36** and **37**.

Compound **36**: 13.1 mg (36%), white solid; R_f_ = 0.16 (1:2 EA/PE); ^1^H NMR (500 MHz, CDCl_3_): δ 7.52 (s, 1H, H-8), 5.44 (t, J = 7.1 Hz, 1H, CH=), 5.15–5.01 (m, 3H, 3 × CH=), 4.91 (d, J = 7.2 Hz, 2H, NCH_2_), 3.58 (s, 3H, N(3)CH_3_), 3.41 (s, 3H, N(1)CH_3_), 2.24–2.19 (m, 2H, CH_2_), 2.15–2.10 (m, 2H, CH_2_), 2.08–2.02 (m, 4H, 2 × CH_2_), 2.00–1.94 (m, 4H, 2 × CH_2_), 1.80 (s, 3H, CH_3_), 1.67 (s, 3H, CH_3_), 1.60 (s, 3H, CH_3_), 1.59 (s, 6H, 2 × CH_3_); ^13^C{^1^H} NMR (150 MHz, CDCl_3_): δ 155.4 (C, C-6), 151.9 (C, C-2), 148.9 (C, C-4), 143.3 (C=), 140.3 (CH, C-8), 136.4 (C=), 135.3 (C=), 131.4 (C=), 124.5 (CH=), 124.1 (CH=), 123.1 (CH=), 118.4 (CH=), 107.2 (C, C-5), 44.5 (NCH_2_), 39.9 (CH_2_), 39.8 (CH_2_), 32.3 (CH_2_), 29.9 (N(3)CH_3_), 28.1 (N(1)CH_3_), 26.9 (CH_2_), 26.7 (CH_2_), 26.5 (CH_2_), 25.9 (CH_3_), 23.6 (CH_3_), 17.8 (CH_3_), 16.21 (CH_3_), 16.15 (CH_3_); IR (film from CH_2_Cl_2_): ν_max_ 3111, 2917, 2853, 1704, 1658 cm^−1^; HRESIMS *m*/*z*: [M + H]^+^ Calcd. for C_27_H_41_N_4_O_2_ 453.3224; Found 453.3215 (Δ = −2.0 ppm); HRESIMS/MS (40 eV) *m*/*z* (%): 181.0700 (100), 124.0496 (36).

Compound **37**: 21.1 mg (19%), white solid; R*_f_* = 0.20 (2:3 EA/PE); ^1^H NMR (500 MHz, CDCl_3_): δ 7.52 (s, 1H, H-8), 5.43 (t, *J* = 7.1 Hz, 1H, CH=), 5.12–5.04 (m, 3H, 3 × CH=), 4.92 (d, *J* = 7.2 Hz, 2H, NCH_2_), 3.58 (s, 3H, N(3)CH_3_), 3.40 (s, 3H, N(1)CH_3_), 2.15–2.07 (m, 4H, 2 × CH_2_), 2.07–2.01 (m, 4H, 2 × CH_2_), 1.99–1.93 (m, 4H, 2 × CH_2_), 1.78 (s, 3H, CH_3_), 1.66 (s, 3H, CH_3_), 1.58 (s, 9H, 3 × CH_3_); ^13^C{^1^H} NMR (150 MHz, CDCl_3_): δ 155.4 (C, C-6), 151.8 (C, C-2), 148.9 (C, C-4), 143.5 (C=), 140.3 (CH, C-8), 136.0 (C=), 135.2 (C=), 131.4 (C=), 124.5 (CH=), 124.2 (CH=), 123.4 (CH=), 117.5 (CH=), 107.2 (C, C-5), 44.6 (NCH_2_), 39.8 (CH_2_), 39.8 (CH_2_), 39.6 (CH_2_), 29.9 (N(3)CH_3_), 28.1 (N(1)CH_3_), 26.9 (CH_2_), 26.7 (CH_2_), 26.3 (CH_2_), 25.8 (CH_3_), 17.8 (CH_3_), 16.7 (CH_3_), 16.2 (CH_3_), 16.1 (CH_3_); IR (film from CH_2_Cl_2_): ν_max_ 3110, 2916, 2854, 1704, 1658 cm^−1^; HRESIMS *m*/*z*: [M + H]^+^ Calcd. for C_27_H_41_N_4_O_2_ 453.3224; Found 453.3205 (Δ = −4.2 ppm); HRESIMS/MS (40 eV) *m*/*z* (%): 181.0693 (100), 124.0489 (31).

#### 3.2.18. (*E*)-7-(3,7-Dimethylocta-2,6-dien-1-yl)-3-methyl-2-thioxo-1,2,3,7-tetrahydro-6*H*-purin-6-one (**38**)

2-Mercapto-3-methylhypoxanthine (0.30 mmol, 54.7 mg), K_2_CO_3_ (0.61 mmol, 84.3 mg) and geranyl bromide (0.39 mmol, 84.9 mg) in DMF (2 mL) at 70 °C for 30 h yielded **30**, after modified work up and modified purification-H_2_O (7 mL) was added to the reaction and the resulting isolated precipitate was dissolved in in MeOH and DCM (1:1, 5 mL). Partial evaporation yielded a precipitate isolated by filtration and washed with MeOH, 5.4 mg (6%), off-white powder. ^1^H NMR (600 MHz, CDCl_3_): δ 9.24 (s, 1H, NH), 7.62 (s, 1H, H-8), 5.43 (t, *J* = 7.3 Hz, 1H, CH=), 5.07–5.03 (m, 1H, CH=), 4.92 (d, *J* = 7.3 Hz, 2H, NCH_2_), 3.93 (s, 3H, NCH_3_), 2.16–2.08 (m, 4H, 2 × CH_2_), 1.78 (s, 3H, CH_3_), 1.69 (s, 3H, CH_3_), 1.60 (s, 3H, CH_3_); ^13^C{^1^H} NMR (150 MHz, CDCl_3_): δ 174.3 (C, C-2), 152.4 (C, C-6), 150.9 (C, C-4), 144.5 (C=), 141.2 (CH, C-8), 132.5 (C=), 123.5 (CH=), 116.8 (CH=), 110.9 (C, C-5), 45.0 (NCH_2_), 39.6 (CH_2_), 35.5 (NCH_3_), 26.2 (CH_2_), 25.9 (CH_3_), 17.9 (CH_3_), 16.7 (CH_3_); IR (neat): ν_max_ 3122, 2964, 2912, 2853, 1708 cm^−1^; HRESIMS *m*/*z*: [M + H]^+^ Calcd. for C_16_H_23_N_4_OS 319.1587; Found 319.1591 (Δ = 1.3 ppm); HRESIMS/MS (40 eV) *m*/*z* (%): 183.033 (48), 149.0452 (19), 126.99579 (11), 124.0503 (100), 96.0557 (21), 81.0702 (46).

#### 3.2.19. 3-Methyl-2-thioxo-7-((6*E*)-3,7,11-trimethyldodeca-2,6,10-trien-1-yl)-1,2,3,7-tetrahydro-6*H*-purin-6-one (**39**)

2-Mercapto-3-methylhypoxanthine (0.32 mmol, 57.8 mg), K_2_CO_3_ (0.61 mmol, 84.7 mg) and farnesyl bromide (0.39 mmol, 111 mg) in DMF (2 mL) at 50 °C for 41 h yielded **39**, 3.3 mg (3%), off-white powder. R*_f_* = 0.19 (1:2 EA/PE); 2:1 *E*/*Z*, NMR data for major isomer: ^1^H NMR (600 MHz, CDCl_3_): δ 9.31 (s, 1H, NH), 7.62 (s, 1H, H-8), 5.43 (t, *J* = 7.3 Hz, 1H, CH=), 5.09–5.05 (m, 2H, 2 × CH=), 4.92 (d, *J* = 7.3 Hz, 2H, NCH_2_), 3.92 (s, 3H, NCH_3_), 2.16–2.09 (m, 4H, 2 × CH_2_), 2.06–2.01 (m, 4H, 2 × CH_2_), 1.78 (s, 3H, CH_3_), 1.66 (s, 3H, CH_3_), 1.59 (s, 3H, CH_3_), 1.58 (s, 3H, CH_3_); ^13^C{^1^H} NMR (150 MHz, CDCl_3_): δ 174.3 (C, C-2), 152.4 (C, C-6), 150.9 (C, C-4), 144.5 (C=), 141.1 (CH, C-8), 136.1 (C=), 131.6 (C=), 124.3 (CH=), 123.3 (CH=), 116.8 (CH=), 110.9 (C, C-5), 45.0 (NCH_2_), 39.8 (CH_2_), 39.6 (CH_2_), 35.5 (NCH_3_), 26.8 (CH_2_), 26.2 (CH_2_), 25.9 (CH_3_), 17.8 (CH_3_), 16.7 (CH_3_), 16.2 (CH_3_); IR (film from CH_2_Cl_2_): ν_max_ 3116, 2922, 2854, 1691 cm^−1^; HRESIMS *m*/*z*: [M + H]^+^ Calcd. for C_21_H_31_N_4_OS 387.2213; Found 387.2216 (Δ = 0.8 ppm); HRESIMS/MS (40 eV) *m*/*z* (%): 183.0316 (100), 149.0436 (13), 124.0493 (42).

#### 3.2.20. 3-Methyl-7-((2*Z*,6*E*,10*E*)-3,7,11,15-tetramethylhexadeca-2,6,10,14-tetraen-1-yl)-2-thioxo-1,2,3,7-tetrahydro-6*H*-purin-6-one (**40**) and 3-methyl-7-((2*E*,6*E*,10*E*)-3,7,11,15-tetramethylhexadeca-2,6,10,14-tetraen-1-yl)-2-thioxo-1,2,3,7-tetrahydro-6H-purin-6-one (**41**)

2-Mercapto-3-methylhypoxanthine (0.30 mmol, 54.7 mg), K_2_CO_3_ (1.1 mmol, 150.8 mg) and geranylgeranyl bromide (0.32 mmol, 111.3 mg) in DMF (1 mL) for 24 h yielded **40** and **41**.

Compound **40**: 3.2 mg (9%), white powder; R*_f_* = 0.14 (1:3 EA/PE); ^1^H NMR (500 MHz, CDCl_3_): δ 9.32 (s, 1H, NH), 7.64 (s, 1H, H-8), 5.45 (t, *J* = 7.2 Hz, 1H, CH=), 5.13–5.07 (m, 3H, 3 × CH=), 4.92 (d, *J* = 7.3 Hz, 2H, NCH_2_), 3.94 (s, 3H, NCH_3_), 2.26–2.19 (m, 2H, CH_2_), 2.18–2.11 (m, 2H, CH_2_), 2.10–2.04 (m, 4H, 2 × CH_2_), 2.02–1.96 (m, 4H, 2 × CH_2_), 1.83 (s, 3H, CH_3_), 1.69 (s, 3H, CH_3_), 1.62 (s, 3H, CH_3_), 1.61 (s, 3H, CH_3_), 1.60 (s, 3H, CH_3_); ^13^C{^1^H} NMR (150 MHz, CDCl_3_): δ 174.3 (C, C-2), 152.4 (C, C-6), 150.9 (C, C-4), 144.2 (C=), 141.2 (CH, C-8), 136.6 (C=), 135.3 (C=), 131.5 (C=), 124.5 (CH=), 124.1 (CH=), 123.0 (CH=), 117.7 (CH=), 110.9 (C, C-5), 44.8 (NCH_2_), 39.9 (CH_2_), 39.8 (CH_2_), 35.5 (NCH_3_), 32.3 (CH_2_), 26.9 (CH_2_), 26.7 (CH_2_), 26.4 (CH_2_), 25.9 (CH_3_), 23.6 (CH_3_), 17.8 (CH_3_), 16.22 (CH_3_), 16.16 (CH_3_); IR (film from CH_2_Cl_2_): ν_max_ 3205, 3118, 2965, 2917, 2855, 1696 cm^−1^; HRESIMS *m*/*z*: [M + H]^+^ Calcd. for C_26_H_39_N_4_OS 455.2839; Found 455.2851 (Δ = 2.6 ppm); HRESIMS/MS (40 eV) *m*/*z* (%): 183.0304 (100), 124.0487 (19).

Compound **41**: 3.4 mg (3%), white powder; R*_f_* = 0.10 (1:3 EA/PE); ^1^H NMR (500 MHz, CDCl_3_): δ 9.29 (s, 1H, NH), 7.62 (s, 1H, H-8), 5.43 (t, *J* = 7.0 Hz, 1H, CH=), 5.13–5.05 (m, 3H, 3 × CH=), 4.92 (d, *J* = 7.3 Hz, 2H, NCH_2_), 3.92 (s, 3H, NCH_3_), 2.17–2.10 (m, 4H, 2 × CH_2_), 2.09–2.02 (m, 4H, 2 × CH_2_), 2.01–1.93 (m, 4H, 2 × CH_2_), 1.79 (s, 3H, CH_3_), 1.68 (s, 3H, CH_3_), 1.60–1.58 (m, 9H, 3 × CH_3_); ^13^C{^1^H} NMR (150 MHz, CDCl_3_): δ 174.3 (C, C-2), 152.4 (C, C-6), 150.9 (C, C-4), 144.5 (C=), 141.1 (CH, C-8), 136.1 (C=), 135.2 (C=), 131.5 (C=), 124.5 (CH=), 124.2 (CH=), 123.3 (CH=), 116.8 (CH=), 110.9 (C, C-5), 45.0 (NCH_2_), 39.9 (CH_2_), 39.8 (CH_2_), 39.6 (CH_2_), 35.5 (NCH_3_), 26.9 (CH_2_), 26.7 (CH_2_), 26.2 (CH_2_), 25.9 (CH_3_), 17.8 (CH_3_), 16.7 (CH_3_), 16.23 (CH_3_), 16.16 (CH_3_); IR (film from CH_2_Cl_2_): ν_max_ 3212, 3117, 2964, 2917, 2853, 1691 cm^−1^; HRESIMS *m*/*z*: [M + H]^+^ Calcd. for C_26_H_39_N_4_OS 455.2839; Found 455.2857 (Δ = 4.0 ppm); HRESIMS/MS (40 eV) *m*/*z* (%): 183.0306 (100), 124.0487 (13).

#### 3.2.21. (*E*)-5-(3,7-Dimethylocta-2,6-dien-1-yl)-1,3-dimethyl-5,7-dihydro-1*H*-purine-2,6,8(3*H*)-trione (**42**)

1,3-Dimethyluric acid (0.12 mmol, 22.6 mg), K_2_CO_3_ (0.119 mmol, 16.5 mg) and geranyl bromide (0.13 mmol, 27.6 mg) in DMF (2 mL) for 24 h yielded **42**, 3.5 mg (9%), white solid. R*_f_* = 0.23 (1:1 EA/PE); ^1^H NMR (600 MHz, CDCl_3_): δ 6.07 (s, 1H, NH), 5.06–5.02 (m, 1H, CH=), 4.95 (t, *J* = 8.1 Hz, 1H, CH=), 3.49 (s, 3H, N(1)CH_3_), 3.28 (s, 3H, N(3)CH_3_), 2.67 (d, *J* = 8.0 Hz, 2H, NCH_2_), 2.12–1.99 (m, 4H, 2 × CH_2_), 1.70 (s, 3H, CH_3_), 1.61 (s, 3H, CH_3_), 1.58 (s, 3H, CH_3_); ^13^C{^1^H} NMR (150 MHz, CDCl_3_): δ 176.9 (C, C-6), 166.4 (C, C-4), 165.4 (C, C-8), 150.8 (C, C-2), 145.6 (C=), 132.6 (C=), 123.6 (CH=), 112.8 (CH=), 68.1 (C, C-5), 40.6 (NCH_2_), 39.9 (CH_2_), 32.0 (N(1)CH_3_), 29.2 (N(3)CH_3_), 26.3 (CH_2_), 25.8 (CH_3_), 17.9 (CH_3_), 16.4 (CH_3_); IR (film from CH_2_Cl_2_): ν_max_ 3278, 3106, 2966, 2921, 2857, 1750, 1697, 1609 cm^−1^; HRESIMS *m*/*z*: [M + H]^+^ Calcd. for C_17_H_25_N_4_O_3_ 333.1910; Found 333.1921 (Δ = 3.3 ppm); HRESIMS/MS (40 eV) *m*/*z* (%): 197.0664 (95), 169.0712 (100), 140.0463 (32), 112.0505 (41).

#### 3.2.22. 1,3-Dimethyl-5-((6*E*)-3,7,11-trimethyldodeca-2,6,10-trien-1-yl)-5,7-dihydro-1H-purine-2,6,8(3*H*)-trione (**43**) and 1,3-dimethyl-9-((6*E*)-3,7,11-trimethyldodeca-2,6,10-trien-1-yl)-7,9-dihydro-1*H*-purine-2,6,8(3*H*)-trione (**44**)

1,3-Dimethyluric acid (0.12 mmol, 23.3 mg), K_2_CO_3_ (0.12 mmol,16.6 mg) and farnesyl bromide (0.11 mmol, 32.0 mg) in DMF (2 mL) for 19 h yielded **43** and **44**.

Compound **43**: 19.2 mg (40%), white solid; R*_f_* = 0.38 (1:1 EA/PE); 2:1 *E*/*Z* NMR data for major isomer: ^1^H NMR (600 MHz, CDCl_3_): δ 6.39 (s, 1H, NH), 5.09–5.02 (m, 2H, 2 × CH=), 4.95 (t, *J* = 7.9 Hz, 1H, CH=), 3.48 (s, 3H, N(1)CH_3_), 3.27 (s, 3H, N(3)CH_3_), 2.68 (d, *J* = 8.1 Hz, 2H, NCH_2_), 2.09–2.00 (m, 6H, 3 × CH_2_), 2.00–1.93 (m, 2H, CH_2_), 1.66 (s, 3H, CH_3_), 1.59 (s, 6H, 2 × CH_3_), 1.58 (s, 3H, CH_3_); ^13^C{^1^H} NMR (150 MHz, CDCl_3_): δ 176.8 (C, C-6), 166.4 (C, C-4), 165.6 (C, C-8), 150.9 (C, C-2), 145.7 (C=), 136.0 (C=), 131.5 (C=), 124.4 (CH=), 123.4 (CH=), 112.7 (CH=), 68.2 (C, C-5), 40.5 (NCH_2_), 40.0 (CH_2_), 39.8 (CH_2_), 32.0 (N(1)CH_3_), 29.2 (N(3)CH_3_), 26.8 (CH_2_), 26.3 (CH_2_), 25.8 (CH_3_), 17.8 (CH_3_), 16.4 (CH_3_), 16.2 (CH_3_); IR (film from CH_2_Cl_2_): ν_max_ 3307, 3098, 2964, 2924, 2855, 1695, 1645, 1612 cm^−1^; HRESIMS *m*/*z*: [M + H]^+^ Calcd. for C_22_H_33_N_4_O_3_ 401.2547; Found 401.2538 (Δ = −2.2 ppm); HRESIMS/MS (40 eV) *m*/*z* (%): 197.0639 (100), 169.0699 (40).

Compound **44**: 3.3 mg (7%), white solid; R*_f_* = 0.13 (1:1 EA/PE); 3:2 *E*/*Z*, NMR data for major isomer: ^1^H NMR (600 MHz, CDCl_3_): δ 8.95 (br s, 1H, NH), 5.16–5.11 (m, 1H, CH=), 5.11–5.00 (m, 2H, 2 × CH=), 4.66 (d, *J* = 5.8 Hz, 2H, NCH_2_), 3.67 (s, 3H, N(3)CH_3_), 3.40 (d, *J* = 2.3 Hz, 3H, N(1)CH_3_), 2.13–2.07 (m, 2H, CH_2_), 2.07–1.98 (m, 4H, 2 × CH_2_), 1.97–1.91 (m, 2H, CH_2_), 1.75 (s, 3H, CH_3_), 1.67 (s, 3H, CH_3_), 1.58 (s, 6H, 2 × CH_3_); ^13^C{^1^H} NMR (150 MHz, CDCl_3_): δ 153.2 (C, C-6), 151.8 (C, C-8), 151.0 (C, C-2), 140.9 (C=), 136.2 (C=), 136.0 (C, C-4), 131.6 (C=), 124.3 (CH=), 123.3 (CH=), 119.5 (CH=), 98.4 (C, C-5), 41.6 (NCH_2_), 39.8 (CH_2_), 39.4 (CH_2_), 31.3 (N(3)CH_3_), 28.6 (N(1)CH_3_), 26.8 (CH_2_), 26.3 (CH_2_), 25.9 (CH_3_), 17.8 (CH_3_), 17.0 (CH_3_), 16.2 (CH_3_); IR (film from CH_2_Cl_2_): ν_max_ 3487, 3174, 3078, 2918, 2854, 1687, 1651 cm^−1^; HRESIMS *m*/*z*: [M + H]^+^ Calcd. for C_22_H_33_N_4_O_3_ 401.2547; Found 401.2551 (Δ = 1.0 ppm); HRESIMS/MS (40 eV) *m*/*z* (%): 197.0658 (100), 169.0707 (57). 

#### 3.2.23. (*E*)-5-(3,7-Dimethylocta-2,6-dien-1-yl)-5*H*-pyrazolo[3,4-*d*]pyrimidin-4-ol (**46**)

Allopurinol (0.80 mmol, 109.2 mg), K_2_CO_3_/Na_2_CO_3_ (1:1, 150.8 mg) and geranyl bromide (0.50 mmol, 108 mg) in DMF (10 mL) for 24 h yielded **46**-additionally recrystallised from MeOH after chromatography, 4.5 mg (2%), white powder. R*_f_* = 0.06 (1:4 EA/PE); ^1^H NMR (600 MHz, CDCl_3_): δ 11.32 (br s, 1H, OH), 8.18 (s, 1H, H-3), 8.02 (s, 1H, H-6), 5.34–5.26 (m, 1H, CH=), 5.07–5.02 (m, 1H, CH=), 4.64 (d, *J* = 7.2 Hz, 2H, NCH_2_), 2.16–2.03 (m, 4H, 2 × CH_2_), 1.82 (s, 3H, CH_3_), 1.67 (s, 3H, CH_3_), 1.59 (s, 3H, CH_3_); ^13^C{^1^H} NMR (150 MHz, CDCl_3_): δ 157.4 (C, C-4), 153.4 (C, C-7a), 149.4 (CH, C-6), 143.0 (C=), 136.4 (CH, C-3), 132.3 (C=), 123.6 (CH=), 118.1 (CH=), 105.9 (C, C-3a), 43.5 (NCH_2_), 39.6 (CH_2_), 26.3 (CH_2_), 25.8 (CH_3_), 17.9 (CH_3_), 16.7 (CH_3_); IR (film from CH_2_Cl_2_): ν_max_ 3188, 3080, 2967, 2905, 2791, 1678, 1568 cm^−1^; HRESIMS *m*/*z*: [M + H]^+^ Calcd. for C_15_H_21_N_4_O 273.1710; Found 273.1715 (Δ = 1.8 ppm); HRESIMS/MS (40 eV) *m*/*z* (%): 137.0454 (100), 110.0349 (39).

#### 3.2.24. 1,5-Bis((6*E*)-3,7,11-trimethyldodeca-2,6,10-trien-1-yl)-1,5-dihydro-4*H*-pyrazolo[3,4-*d*]pyrimidin-4-one (**47**), 2,7-bis((6*E*)-3,7,11-trimethyldodeca-2,6,10-trien-1-yl)-2,7-dihydro-4*H*-pyrazolo[3,4-*d*]pyrimidin-4-one (**48**), and 5-((2*E*,6*E*)-3,7,11-trimethyldodeca-2,6,10-trien-1-yl)-5*H*-pyrazolo[3,4-*d*]pyrimidin-4-ol (**49**)

Allopurinol (0.48 mmol, 65.2 mg), K_2_CO_3_ (0.51 mmol, 70.5 mg) and farnesyl bromide (0.55 mmol, 157.0 mg) in DMF (2 mL) at 70 °C for 27 h yielded **47**, **48**, and **49**.

Compound **47**: 17.2 mg (12%), colourless oil; R*_f_* = 0.24 (1:5 EA/PE); 2:1 *E*/*Z*, NMR data for major isomer: ^1^H NMR (500 MHz, CDCl_3_): δ 8.09–8.08 (m, 1H, H-6), 7.95–7.94 (m, 1H, H-3), 5.48–5.41 (m, 1H, CH=), 5.34–5.28 (m, 1H, CH=), 5.13–5.04 (m, 4H, 4 × CH=), 4.94 (d, *J* = 6.9 Hz, 2H, NCH_2_), 4.61 (d, *J* = 7.2 Hz, 2H, NCH_2_), 2.18–1.99 (m, 14H, 7 × CH_2_), 1.99–1.92 (m, 2H, CH_2_), 1.85 (s, 3H, CH_3_), 1.83 (s, 3H, CH_3_), 1.68 (s, 9H, 3 × CH_3_), 1.60 (s, 9H, 3 × CH_3_); ^13^C{^1^H} NMR (150 MHz, CDCl_3_): δ 157.4 (C, C-4), 151.0 (C, C-7a), 148.4 (CH, C-6), 142.7 (C=), 141.0 (C=), 135.9 (C=), 135.6 (C=), 135.1 (CH, C-3), 131.49 (C=), 131.46 (C=), 124.42 (CH=), 124.37 (CH=), 123.7 (CH=), 123.5 (CH=), 118.4 (CH=), 118.3 (CH=), 105.9 (C, C-3a), 45.4 (NCH_2_), 43.2 (NCH_2_), 39.80 (CH_2_), 39.78 (CH_2_), 39.64 (CH_2_), 39.59 (CH_2_), 26.83 (CH_2_), 26.81 (CH_2_), 26.34 (CH_2_), 26.30 (CH_2_), 25.9 (CH_3_), 25.8 (CH_3_), 17.8 (2 × CH_3_), 16.74 (CH_3_), 16.69 (CH_3_), 16.19 (CH_3_), 16.15 (CH_3_); IR (film from CH_2_Cl_2_): ν_max_ 3368, 2964, 2925, 2856, 1696, 1582 cm^−1^; HRESIMS *m*/*z*: [M + H]^+^ Calcd. for C_35_H_53_N_4_O 545.4214; Found 545.4223 (Δ = 1.7 ppm); HRESIMS/MS (40 eV) *m*/*z* (%): 137.0458 (100), 81.702 (12).

Compound **48**: 6.4 mg (4%), colourless oil; R*_f_* = 0.48 (1:1 EA/PE); 3:2 *E*/*Z*, NMR data for major isomer: ^1^H NMR (500 MHz, CDCl_3_): δ 8.06 (s, 1H, H-3), 7.95 (s, 1H, H-6), 5.51 (t, *J* = 6.5 Hz, 1H, CH=), 5.32–5.26 (m, 1H, CH=), 5.13–5.05 (m, 4H, 4 × CH=), 4.89 (d, *J* = 7.4 Hz, 2H, NCH_2_), 4.57 (d, *J* = 7.1 Hz, 2H, NCH_2_), 2.20–1.93 (m, 16H, 8 × CH_2_), 1.81 (s, 3H, CH_3_), 1.79 (s, 3H, CH_3_), 1.68 (s, 6H, 2 × CH_3_), 1.62–1.57 (m, 12H, 4 × CH_3_); ^13^C{^1^H} NMR (150 MHz, CDCl_3_): δ 158.9 (C, C-4), 158.6 (C, 7a), 148.5 (CH, C-6), 144.5 (C=), 142.1 (C=), 136.0 (C=), 135.8 (C=), 131.7 (C=), 131.4 (C=), 127.0 (CH, C-3), 124.4 (CH=), 124.3 (CH=), 123.5 (CH=), 123.4 (CH=), 118.5 (CH=), 116.7 (CH=), 107.1 (C, C-3a), 51.2 (NCH_2_), 43.0 (NCH_2_), 39.77 (CH_2_), 39.76 (CH_2_), 39.64 (CH_2_), 39.62 (CH_2_), 26.8 (CH_2_), 26.7 (CH_2_), 26.4 (CH_2_), 26.3 (CH_2_), 25.85 (CH_3_), 25.84 (CH_3_), 17.82 (CH_3_), 17.81 (CH_3_), 16.8 (CH_3_), 16.7 (CH_3_), 16.19 (CH_3_), 16.15 (CH_3_); IR (film from CH_2_Cl_2_): ν_max_ 3404, 2973, 2934, 1687 cm^−1^; HRESIMS *m*/*z*: [M + H]^+^ Calcd. for C_35_H_53_N_4_O 545.4214; Found 545.4225 (Δ = 2.0 ppm); HRESIMS/MS (40 eV) *m*/*z* (%): 137.0451 (100), 81.0699 (30).

Compound **49**: 33.6 mg (21%), white solid; R*_f_* = 0.16 (1:1 EA/PE); 3:2 *E*/*Z*, NMR data for major isomer: ^1^H NMR (600 MHz, CDCl_3_): δ 12.64 (br s, 1H, OH), 8.20 (s, 1H, H-3), 8.08 (s, 1H, H-6), 5.34–5.28 (m, 1H, CH=), 5.09–5.01 (m, 2H, 2 × CH=), 4.64 (d, *J* = 7.3 Hz, 2H, NCH_2_), 2.14–2.04 (m, 4H, 2 × CH_2_), 2.04–1.96 (m, 2H, CH_2_), 1.96–1.92 (m, 2H, CH_2_), 1.82 (s, 3H, CH_3_), 1.64 (s, 3H, CH_3_), 1.57 (s, 3H, CH_3_), 1.56 (s, 3H, CH_3_); ^13^C{^1^H} NMR (150 MHz, CDCl_3_): δ 157.4 (C, C-4), 153.3 (C, C-7a), 149.4 (CH, C-6), 143.0 (C=), 136.1 (CH, C-3) 135.9 (C=), 131.5 (C=), 124.3 (CH=), 123.4 (CH=), 118.0 (CH=), 105.9 (C, C-3a), 43.5 (NCH_2_), 39.7 (CH_2_), 39.6 (CH_2_), 26.8 (CH_2_), 26.3 (CH_2_), 25.8 (CH_3_), 17.8 (CH_3_), 16.7 (CH_3_), 16.2 (CH_3_); IR (film from CH_2_Cl_2_): ν_max_ 3188, 3108, 2967, 2917, 1676 cm^−1^; HRESIMS *m*/*z*: [M + H]^+^ Calcd. for C_20_H_29_N_4_O 341.2336; Found 341.2340 (Δ = 1.2 ppm); HRESIMS/MS (40 eV) *m*/*z* (%): 137.0451 (100), 110.0344 (9).

#### 3.2.25. 1,5-Bis((2*E*,6*E*,10*E*)-3,7,11,15-tetramethylhexadeca-2,6,10,14-tetraen-1-yl)-1,5-dihydro-4*H*-pyrazolo[3,4-*d*]pyrimidin-4-one (**50**)

Allopurinol (0.71 mmol, 96.6 mg), K_2_CO_3_ (0.81 mmol, 111.9 mg) and geranylgeranyl bromide (0.73 mmol, 259.7 mg) in DMF (1 mL) for 27 h yielded **50**, 6.8 mg (2%), colourless oil. R*_f_* = 0.26 (1:5 EA/PE); ^1^H NMR (600 MHz, CDCl_3_): δ 8.07 (s, 1H, H-3), 7.93 (s, 1H, H-6), 5.43 (t, *J* = 6.8 Hz, 1H, CH=), 5.30 (t, *J* = 7.2 Hz, 1H, CH=), 5.11–5.04 (m, 6H, 6 × CH=), 4.93 (d, *J* = 6.9 Hz, 2H, NCH_2_), 4.60 (d, *J* = 7.3 Hz, 2H, NCH_2_), 2.14–2.01 (m, 16H, 8 × CH_2_), 1.99–1.93 (m, 8H, 4 × CH_2_), 1.83 (s, 3H, CH_3_), 1.81 (s, 3H, CH_3_), 1.67 (s, 6H, 2 × CH_3_), 1.59 (s, 6H, 2 × CH_3_), 1.584 (s, 3H, CH_3_), 1.579 (s, 6H, 2 × CH_3_), 1.56 (s, 3H, CH_3_); ^13^C{^1^H} NMR (150 MHz, CDCl_3_): δ 157.4 (C, C-4), 151.0 (C, C-7a), 148.4 (CH, C-6), 142.7 (C=), 141.1 (C=), 135.9 (C=), 135.6 (C=), 135.13 (C= and CH, C-3), 135.09 (C=), 131.43 (C=), 131.41 (C=), 124.50 (CH=), 124.48 (CH=), 124.3 (CH=), 124.2 (CH=), 123.7 (CH=), 123.5 (CH=), 118.4 (CH=), 118.3 (CH=), 105.9 (C, C-3a), 45.4 (NCH_2_), 43.2 (NCH_2_), 39.86 (CH_2_), 39.85 (CH_2_), 39.80 (CH_2_), 39.78 (CH_2_), 39.7 (CH_2_), 39.6 (CH_2_), 26.9 (2 × CH_2_), 26.73 (CH_2_), 26.71 (CH_2_), 26.38 (CH_2_), 26.35 (CH_2_), 25.9 (2 × CH_3_), 17.8 (2 × CH_3_), 16.8 (CH_3_), 16.7 (CH_3_), 16.2 (2 × CH_3_), 16.1 (2 × CH_3_); IR (film from CH_2_Cl_2_): ν_max_ 3392, 2975, 2937, 1699 cm^−1^; HRESIMS *m*/*z*: [M + H]^+^ Calcd. for C_45_H_69_N_4_O 681.5466; Found 681.5470 (Δ = 0.6 ppm); HRESIMS/MS (40 eV) *m*/*z* (%): 341.01647 (72), 281.0500 (20), 266.9983 (21), 221.0823 (24), 207.0312 (25), 147.0648 (32), 137.0451 (100).

#### 3.2.26. (*E*)-1-(3,7-Dimethylocta-2,6-dien-1-yl)pyrimidine-2,4(1*H*,3*H*)-dione (**54**)

Uracil (0.53 mmol, 59.9 mg), K_2_CO_3_ (0.51 mmol, 68.9 mg) and geranyl bromide (0.65 mmol, 141.1 mg) in DMF (2 mL) at 50 °C for 21 h yielded **54**, after modified work up and modified purification—the reaction was filtered, H_2_O (6 mL) was added to the filtrate and stored in the fridge until precipitate formed. The isolated solid was recrystallised from PE, 15.8 mg (12%), white crystals. ^1^H NMR (500 MHz, CDCl_3_): δ 8.42 (br s, 1H, NH), 7.16 (d, *J* = 7.9 Hz, 1H, H-6), 5.68 (dd, *J* = 7.9, 2.1 Hz, 1H, H-5), 5.22 (t, *J* = 7.3, 1H, CH=), 5.04 (t, *J* = 6.6 Hz, 1H, CH=), 4.35 (d, *J* = 7.3 Hz, 2H, NCH_2_), 2.11 (m, 4H, 2 × CH_2_), 1.75 (s, 3H, CH_3_), 1.68 (s, 3H, CH_3_), 1.60 (s, 3H, CH_3_); ^13^C{^1^H} NMR (150 MHz, CDCl_3_): δ 163.4 (C, C-4), 150.9 (C, C-2), 143.7 (C=), 143.6 (CH, C-6), 132.4 (C=), 123.5 (CH=), 117.3 (CH=), 102.2 (CH, C-5), 45.2 (NCH_2_), 39.6 (CH_2_), 26.2 (CH_2_), 25.9 (CH_3_), 17.9 (CH_3_), 16.6 (CH_3_); IR (neat): ν_max_ 3121, 2967, 2929, 2808, 1696, 1657 cm^−1^; HRESIMS *m*/*z*: [M + H]^+^ Calcd. for C_14_H_21_N_2_O_2_ 249.1598; Found 249.1595 (Δ = −1.2 ppm); HRESIMS/MS (40 eV) *m*/*z* (%): 113.0346 (100), 81.0700 (18), 70.0289 (7).

#### 3.2.27. 1-((6*E*)-3,7,11-Trimethyldodeca-2,6,10-trien-1-yl)pyrimidine-2,4(1*H*,3*H*)-dione (**55**)

Uracil (0.49 mmol, 54.9 mg), K_2_CO_3_ (0.51 mmol, 69.9 mg) and farnesyl bromide (0.54 mmol, 150 mg) in DMF (2 mL) at 70 °C for 24 h yielded **55**, after modified work up and modified purification as per **54**, 38.6 mg (25%), white crystals. 3:2 *E*/*Z*, NMR data for major isomer: ^1^H NMR (600 MHz, CDCl_3_): δ 8.46 (s, 1H, NH), 7.16 (d, *J* = 7.9 Hz, 1H, H-6), 5.68 (dd *J* = 8.0, 2.2 Hz, 1H, H-5), 5.24–5.19 (m, 1H, CH=), 5.09–5.04 (m, 2H, 2 × CH=), 4.34 (d, *J* = 7.3 Hz, 2H, NCH_2_), 2.16–2.01 (m, 6H, 3 × CH_2_), 1.99–1.94 (m, 2H, CH_2_), 1.75 (s, 3H, CH_3_), 1.68 (s, 3H, CH_3_), 1.62 (s, 3H, CH_3_), 1.59 (s, 3H, CH_3_); ^13^C{^1^H} NMR (150 MHz, CDCl_3_): δ 163.5 (C, C-4), 150.9 (C, C-2), 143.8 (C=), 143.6 (CH, C-6), 136.0 (C=), 131.6 (C=), 124.3 (CH=), 123.4 (CH=), 117.2 (CH=), 102.2 (CH, C-5), 45.3 (NCH_2_), 39.8 (CH_2_), 39.6 (CH_2_), 26.8 (CH_2_), 26.2 (CH_2_), 25.9 (CH_3_), 17.8 (CH_3_), 16.6 (CH_3_), 16.2 (CH_3_); IR (film from CH_2_Cl_2_): ν_max_ 3054, 2979, 2930, 1683 cm^−1^; HRESIMS *m*/*z*: [M + H]^+^ Calcd. for C_19_H_29_N_2_O_2_ 317.2224; Found 317.2222 (Δ = −0.6 ppm); HRESIMS/MS (40 eV) *m*/*z* (%): 113.0340 (87), 81.0698 (100).

#### 3.2.28. 1-((2*Z*,6*E*,10*E*)-3,7,11,15-Tetramethylhexadeca-2,6,10,14-tetraen-1-yl)pyrimidine-2,4(1*H*,3*H*)-dione (**56**)

Uracil (0.29 mmol, 32.7 mg), K_2_CO_3_ (0.36 mmol, 50.1 mg) and geranylgeranyl bromide (0.32 mmol, 111 mg) in DMF (1 mL) for 48 h yielded **56**, 9.4 mg (31%), white solid. R*_f_* = 0.23 (2:3 EA/PE); ^1^H NMR (500 MHz, CDCl_3_): δ 8.78 (s, 1H, NH), 7.15 (dd, *J* = 7.9, 0.8 Hz, 1H, H-6), 5.68 (dd, *J* = 7.9, 1.4 Hz, 1H, H-5), 5.22 (t, *J* = 7.2 Hz, 1H, CH=), 5.13–5.04 (m, 3H, 3 × CH=), 4.33 (d, *J* = 7.3 Hz, 2H, NCH_2_), 2.20–2.10 (m, 4H, 2 × CH_2_), 2.10–2.02 (m, 4H, 2 × CH_2_), 2.01–1.93 (m, 4H, 2 × CH_2_), 1.79 (s, 3H, CH_3_), 1.68 (s, 3H, CH_3_), 1.61 (s, 3H, CH_3_), 1.60 (s, 6H, 2 × CH_3_); ^13^C{^1^H} NMR (150 MHz, CDCl_3_): δ 163.6 (C, C-4), 150.9 (C, C-2), 143.7 (C=), 143.6 (CH, C-6), 136.5 (C=), 135.3 (C=), 131.5 (C=), 124.5 (CH=), 124.1 (CH=), 123.1 (CH=), 118.1 (CH=), 102.2 (CH, C-5), 45.1 (NCH_2_), 39.9 (2 × CH_2_), 32.2 (CH_2_), 26.9 (CH_2_), 26.7 (CH_2_), 26.5 (CH_2_), 25.9 (CH_3_), 23.6 (CH_3_), 17.8 (CH_3_), 16.20 (CH_3_), 16.16 (CH_3_); IR (film from CH_2_Cl_2_): ν_max_ 3425, 3197, 3054, 2970, 2930, 2875, 1686 cm^−1^; HRESIMS *m*/*z*: [M + H]^+^ Calcd. C_24_H_37_N_2_O_2_ 385.2850; Found 385.2817 (Δ = −8.6 ppm); HRESIMS/MS (40 eV) *m*/*z* (%):113.0331 (100), 107.0389 (21).

#### 3.2.29. (*E*)-1-(3,7-Dimethylocta-2,6-dien-1-yl)-5-methylpyrimidine-2,4-(1*H*,3*H*)-dione (**57**)

Thymine (0.48 mmol, 61.0 mg), K_2_CO_3_ (0.50 mmol, 69.4 mg) and geranyl bromide (0.65 mmol, 141.1 mg) in DMF (2 mL) for 20 h yielded 57, after modified work up and modified purification as per 54, 6.8 mg (5%), white crystals. ^1^H NMR (600 MHz, CDCl_3_): δ 8.20 (br s, 1H, NH), 6.95 (d, *J* = 1.2 Hz, 1H, H-6), 5.21 (t, *J* = 7.1 Hz, 1H, CH=), 5.05 (t, *J* = 6.9 Hz, 1H, CH=), 4.32 (d, *J* = 7.3 Hz, 2H, NCH_2_), 2.14–2.05 (m, 4H, 2 × CH_2_), 1.92 (s, 3H,C(5)CH_3_), 1.75 (s, 3H, CH_3_), 1.67 (s, 3H, CH_3_), 1.60 (s, 3H, CH_3_); ^13^C{^1^H} NMR (150 MHz, CDCl_3_): δ 163.9 (C, C-4), 150.8 (C, C-2), 142.9 (C=), 139.7 (CH, C-6), 132.3 (C=), 123.5 (CH=), 117.7 (CH=), 110.7 (C, C-5), 45.1 (NCH_2_), 39.6 (CH_2_), 26.3 (CH_2_), 25.9 (CH_3_), 17.9 (CH_3_), 16.6 (CH_3_), 12.6 (C(5)CH_3_); IR (film from CH_2_Cl_2_): ν_max_ 3152, 2975, 2919, 2830, 1685, 1645 cm^−1^; HRESIMS *m*/*z*: [M + H]^+^ Calcd. for C_15_H_23_N_2_O_2_ 263.1754; Found 263.1752 (Δ = −0.8 ppm); HRESIMS/MS (10 eV) *m*/*z* (%): 128.0524 (7), 127.0501 (100), 81.07 (19).

#### 3.2.30. 5-Methyl-1-((6*E*)-3,7,11-trimethyldodeca-2,6,10-trien-1-yl)pyrimidine-2,4(1*H*,3*H*)-dione (**58**)

Thymine (0.49 mmol, 62.4 mg), K_2_CO_3_ (0.51 mmol, 70.5 mg) and farnesyl bromide (0.62 mmol, 176.9 mg) in DMF (2 mL) at 70 °C for 24 h yielded **58**, after modified work up and modified purification as per **54**, 31.9 mg (20%), white crystals. 3:2 *E*/*Z*, NMR data for major isomer: ^1^H NMR (600 MHz, CDCl_3_): δ 8.41 (br s, 1H, NH), 6.95 (s, 1H, H-6), 5.24–5.19 (m, 1H, CH=), 5.11–5.01 (m, 2H, 2 × CH=), 4.32 (d, *J* = 7.5 Hz, 2H, NCH_2_), 2.14–2.02 (m, 6H, 3 × CH_2_), 1.98–1.95 (m, 2H, CH_2_), 1.91 (s, 3H, C(5)CH_3_), 1.76 (s, 3H, CH_3_), 1.67 (s, 6H, 2 × CH_3_), 1.59 (s, 3H, CH_3_); ^13^C{^1^H} NMR (150 MHz, CDCl_3_): δ 164.0 (C, C-4), 150.9 (C, C-2), 142.9 (C=), 139.7 (CH, C-6), 135.9 (C=), 131.6 (C=), 124.3 (CH=), 123.4 (CH=), 117.7 (CH=), 110.7 (C, C-5), 45.1 (NCH_2_), 39.8 (CH_2_), 39.6 (CH_2_), 26.8 (CH_2_), 26.2 (CH_2_), 25.9 (CH_3_), 17.8 (CH_3_), 16.7 (CH_3_), 16.2 (CH_3_), 12.6 (C(5)CH_3_); IR (film from CH_2_Cl_2_): ν_max_ 3177, 3052, 2966, 2927, 1665 cm^−1^; HRESIMS *m*/*z*: [M + H]^+^ Calcd. for C_20_H_31_N_2_O_2_ 331.2380; Found 331.2372 (Δ = −2.4 ppm); HRESIMS/MS (40 eV) *m*/*z* (%): 127.0487 (100), 110.0219 (46).

#### 3.2.31. 5-Methyl-1-((2*E*,6*E*,10*E*)-3,7,11,15-tetramethylhexadeca-2,6,10,14-tetraen-1-yl)pyrimidine-2,4(1*H*,3*H*)-dione (**59**)

Thymine (0.31 mmol, 38.7 mg), K_2_CO_3_ (0.36 mmol, 49.6 mg) and geranylgeranyl bromide (0.32 mmol, 111 mg) in DMF (1 mL) for 24 h yielded **59**, 10.1 mg (11%), waxy white solid. R*_f_* = 0.22 (2:3 EA/PE); ^1^H NMR (500 MHz, CDCl_3_): δ 8.67 (br s, 1H, NH), 6.95 (d, *J* = 1.1 Hz, 1H, H-6), 5.21 (t, *J* = 6.7 Hz, 1H, CH=), 5.12–5.04 (m, 3H, 3 × CH=), 4.32 (d, *J* = 7.1 Hz, 2H, NCH_2_), 2.16–2.02 (m, 8H, 4 × CH_2_), 2.01–1.94 (m, 4H, 2 × CH_2_), 1.91 (s, 3H, C(5)CH_3_), 1.76 (s, 3H, CH_3_), 1.68 (s, 3H, CH_3_), 1.60 (s, 6H, 2 × CH_3_), 1.59 (s, 3H, CH_3_); IR (film from CH_2_Cl_2_): ν_max_ 3427, 3176, 3043, 2968, 2925, 2855, 1668 cm^−1^; ^13^C{^1^H} NMR (150 MHz, CDCl_3_): δ 164.1 (C, C-4), 151.0 (C, C-2), 143.0 (C=), 139.7 (CH, C-6), 135.9 (C=), 135.2 (C=), 131.4 (C=), 124.5 (CH=) 124.2 (CH=), 123.5 (CH=), 117.7 (CH=), 110.8 (C, C-5), 45.1 (NCH_2_), 39.9 (CH_2_), 39.8 (CH_2_), 39.7 (CH_2_), 26.9 (CH_2_), 26.7 (CH_2_), 26.3 (CH_2_), 25.9 (CH_3_), 17.8 (CH_3_), 16.7 (CH_3_), 16.21 (CH_3_), 16.15 (CH_3_), 12.6 (C(5)CH_3_); HRESIMS *m*/*z*: [M + H]^+^ Calcd. for C_25_H_39_N_2_O_2_ 399.3006; Found 399.2995 (Δ = −2.8 ppm); HRESIMS/MS *m*/*z* (20 eV) (%): 399.2980 (8), 283.2625 (13), 127.0501 (100).

#### 3.2.32. *N*-(-1,3-Bis((*E*)-3,7-dimethylocta-2,6-dien-1-yl)-2-oxo-2,3-dihydropyrimidin-4(1*H*)-ylidene)formamide (**60**) and 1,3-bis((*E*)-3,7-dimethylocta-2,6-dien-1-yl)-4-imino-3,4-dihydropyrimidin-2(1*H*)-one (**61**)

Cytosine (1.0 mmol, 112.4 mg), K_2_CO_3_ (1.0 mmol, 152.0 mg) and geranyl bromide (1.1 mmol, 238.5 mg) in DMF (4 mL) for 27 h yielded **60** and **61**.

Compound **60**: 4.5 mg (2%), pale-yellow oil; R*_f_* = 0.40 (1:1 EA/PE); ^1^H NMR (600 MHz, CDCl_3_): δ 9.14 (s, 1H, HC=O), 7.10 (d, *J* = 7.9 Hz, 1H, H-6), 6.4 (d, *J* = 7.9 Hz, 1H, H-5), 5.25 (t, *J* = 7.0 Hz, 1H, CH=), 5.22 (t, *J* = 7.0 Hz, 1H, CH=), 5.08–5.02 (m, 2H, 2 × CH=), 4.75 (d, *J* = 7.1 Hz, 2H, NCH_2_), 4.39 (d, *J* = 7.2 Hz, 2H, NCH_2_), 2.15–2.03 (m, 6H, 3 × CH_2_), 2.02–1.97 (m, 2H, CH_2_), 1.82 (s, 3H, CH_3_), 1.74 (s, 3H, CH_3_), 1.68 (s, 3H, CH_3_), 1.66 (s, 3H, CH_3_), 1.60 (s, 3H, CH_3_), 1.58 (s, 3H, CH_3_); ^13^C{^1^H} NMR (150 MHz, CDCl_3_): δ 172.8 (HC=O), 159.8 (C, C-4), 150.5 (C, C-2), 144.3 (C=), 141.2 (C=), 140.8 (CH, C-6), 132.4 (C=), 131.7 (C=), 124.1 (CH=), 123.5 (CH=), 117.4 (CH=), 117.0 (CH=), 97.4 (CH, C-5), 46.6 (NCH_2_), 42.0 (NCH_2_), 39.8 (CH_2_), 39.6 (CH_2_), 26.6 (CH_2_), 26.2 (CH_2_), 25.89 (CH_3_), 25.85 (CH_3_), 17.89 (CH_3_), 17.85 (CH_3_), 16.8 (CH_3_), 16.7 (CH_3_); IR (film from CH_2_Cl_2_): ν_max_ 3306, 2964, 2915, 2854, 1654, 1684, 1403 cm^−1^; HRESIMS *m*/*z*: [M + H]^+^ Calcd. for C_25_H_38_N_3_O_2_ 412.2959; Found 412.2985 (Δ = 6.3 ppm); HRESIMS/MS *m*/*z* (10 eV) (%): 412.0949 (11), 276.1708 (35), 140.0447 (100).

Compound **61**: 18.3 mg (9%), pale-yellow oil; R*_f_* = 0.19 (1:2 EA/PE); ^1^H NMR (600 MHz, CDCl_3_): δ 6.59 (d, *J* = 7.9 Hz, 1H, H-6), 5.53 (d, *J* = 7.9 Hz, 1H, H-5), 5.23 (t, *J* = 6.3 Hz, 1H, CH=), 5.21–5.16 (m, 1H, CH=), 5.09–5.02 (m, 2H, 2 × CH=), 4.62 (d, *J* = 6.4 Hz, 2H, NCH_2_), 4.25 (d, *J* = 7.2 Hz, 2H, NCH_2_), 2.14–2.03 (m, 6H, 3 × CH_2_), 2.01–1.97 (m, 2H, CH_2_), 1.80 (s, 3H, CH_3_), 1.71 (s, 3H, CH_3_), 1.66 (s, 3H, CH_3_), 1.65 (s, 3H, CH_3_), 1.58 (s, 3H, CH_3_), 1.57 (s, 3H, CH_3_); ^13^C{^1^H} NMR (150 MHz, CDCl_3_): δ 158.2 (C, C-4), 151.3 (C, C-2), 142.3 (C=), 139.6 (C=), 135.2 (CH, C-6), 132.1 (C=), 131.6 (C=), 124.2 (CH=), 123.7 (CH=), 118.6 (CH=), 118.3 (CH=), 102.0 (CH, C-5), 45.7 (NCH_2_), 40.5 (NCH_2_), 39.8 (CH_2_), 39.6 (CH_2_), 26.6 (CH_2_), 26.3 (CH_2_), 25.83 (CH_3_), 25.81 (CH_3_), 17.9 (CH_3_), 17.8 (CH_3_), 16.7 (CH_3_), 16.5 (CH_3_); IR (film from CH_2_Cl_2_): ν_max_ 3305, 3083, 2966, 2915, 2855, 1651 cm^−1^; HRESIMS *m*/*z*: [M + H]^+^ Calcd. for C_24_H_38_N_3_O 384.3009; Found 384.3023 (Δ = 3.6 ppm); HRESIMS/MS *m*/*z* (20 eV) (%): 384.3018 (0.1), 248.1764 (100), 113.0529 (0.09).

#### 3.2.33. *N*-(-2-Oxo-1,3-bis((6*E*)-3,7,11-trimethyldodeca-2,6,10-trien-1-yl)-2,3-dihydropyrimidin-4(1*H*)-ylidene)formamide (**62**) and 4-imino-1,3-bis((6*E*)-3,7,11-trimethyldodeca-2,6,10-trien-1-yl)-3,4-dihydropyrimidin-2(1*H*)-one (**63**)

Cytosine (1.0 mmol, 114.8 mg), K_2_CO_3_ (1.5 mmol, 204.4 mg) and farnesyl bromide (1.2 mmol, 341 mg) in DMF (2 mL) for 25 h yielded **62** and **63**.

Compound **62**: 11.3 mg (3%), pale-yellow oil; R*_f_* = 0.16 (1:4 EA/PE); 3:2 *E*/*Z*, data for major isomer: ^1^H NMR (600 MHz, CDCl_3_): δ 9.13 (s, 1H, HC=O), 7.1 (d, *J* = 7.8 Hz, 1H, H-6), 6.39 (d, *J* = 7.8 Hz, 1H, H-5), 5.28–5.20 (m, 2H, 2 × CH=), 5.12–5.04 (m, 4H, 4 × CH=), 4.75 (d, *J* = 6.8 Hz, 2H, NCH_2_), 4.39 (d, *J* = 7.4 Hz, 2H, NCH_2_), 2.15–1.92 (m, 16H, 8 × CH_2_), 1.83 (s, 3H, CH_3_), 1.75 (s, 3H, CH_3_), 1.67 (s, 6H, 2 × CH_3_), 1.60 (s, 3H, CH_3_), 1.59 (s, 6H, 2 × CH_3_), 1.57 (s, 3H, CH_3_); ^13^C{^1^H} NMR (150 MHz, CDCl_3_): δ 172.8 (C, HC=O), 159.8 (C, C-4), 150.5 (C, C-2), 144.3 (C=), 141.3 (C=), 140.8 (CH, C-6), 136.0 (C=), 135.4 (C=), 131.6 (C=), 131.4 (C=), 124.5 (CH=), 124.3 (CH=), 124.0 (CH=), 123.3 (CH=), 117.4 (CH=), 116.9 (CH=), 97.4 (CH, C-5), 46.6 (NCH_2_), 42.0 (NCH_2_), 39.84 (CH_2_), 39.82 (2 × CH_2_), 39.7 (CH_2_), 26.9 (CH_2_), 26.8 (CH_2_), 26.5 (CH_2_), 26.2 (CH_2_), 25.9 (2 × CH_3_), 17.852 (CH_3_), 17.846 (CH_3_), 16.8 (CH_3_), 16.7 (CH_3_), 16.22 (CH_3_), 16.16 (CH_3_); IR (film from CH_2_Cl_2_): ν_max_ 2964, 2915, 2854, 1650, 1450 cm^−1^; HRESIMS *m*/*z*: [M + H]^+^ Calcd. for C_35_H_54_N_3_O_2_ 548.4211; Found 548.4233 (Δ = 4.0 ppm).

Compound **63**: 96.4 mg (31%), pale-yellow oil; R*_f_* = 0.19 (1:1 EA/PE); 3:2 *E*/*Z*, data for major isomer: ^1^H NMR (600 MHz, CDCl_3_): δ 6.59 (d, *J* = 7.9 Hz, 1H, H-6), 5.53 (d, *J* = 7.7 Hz, 1H, H-5), 5.24 (t, *J* = 5.7 Hz, 1H, CH=), 5.20 (t, *J* = 6.8 Hz, 1H, CH=), 5.13–5.04 (m, 4H, 4 × CH=), 4.63 (d, *J* = 6.3 Hz, 2H, NCH_2_), 4.26 (d, *J* = 7.2 Hz, 2H, NCH_2_), 2.15–1.91 (m, 16H, 8 × CH_2_), 1.81 (s, 3H, CH_3_), 1.72 (s, 3H, CH_3_), 1.68 (s, 9H, 3 × CH_3_), 1.59 (s, 9H, 3 × CH_3_); ^13^C{^1^H} NMR (150 MHz, CDCl_3_): δ 158.2 (C, C-4), 151.4 (C, C-2), 142.3 (C=), 139.6 (C=), 135.7 (C=), 135.1 (C=), 134.9 (CH, C-6), 131.6 (C=), 131.4 (C=), 124.5 (CH=), 124.3 (CH=), 124.1 (CH=), 123.6 (CH=), 118.7 (CH=), 118.4 (CH=), 102.2 (CH, C-5), 45.7 (NCH_2_), 40.4 (NCH_2_), 39.84 (CH_2_), 39.83 (CH_2_), 39.77 (CH_2_), 39.65 (CH_2_), 26.9 (CH_2_), 26.8 (CH_2_), 26.6 (CH_2_), 26.3 (CH_2_), 25.0 (2 × CH_3_), 17.9 (2 × CH_3_), 16.7 (CH_3_), 16.6 (CH_3_), 16.2 (CH_3_), 16.1 (CH_3_); IR (film from CH_2_Cl_2_): ν_max_ 3306, 2964, 2915, 2854 1654 cm^−1^; HRESIMS *m*/*z*: [M + H]^+^ Calcd. for C_34_H_54_N_3_O 520.4261; Found 520.4269 (Δ = 1.5 ppm); HRESIMS/MS *m*/*z* (20 eV) (%): 316.2386 (21), 112.0511 (100).

#### 3.2.34. *N*-(-2-Oxo-1,3-bis((2*E*,6*E*,10*E*)-3,7,11,15-tetramethylhexadeca-2,6,10,14-tetraen-1-yl)-2,3-dihydropyrimidin-4(1*H*)-ylidene)formamide (**64**)

Cytosine (0.41 mmol, 45.7mg), K_2_CO_3_ (0.84 mmol, 116.4 mg), and geranylgeranyl bromide (0.42 mmol, 148.4 mg) in DMF (2 mL) for 25 h yielded **64**, 6.3 mg (3%), pale-yellow oil. R*_f_* = 0.50 (1:1 EA/PE); ^1^H NMR (600 MHz, CDCl_3_): δ 9.13 (s, 1H, HC=O), 7.09 (d, *J* = 7.9 Hz, 1H, H-6), 6.39 (d, *J* = 7.9 Hz, 1H, H-5), 5.26 (t, *J* = 7.0 Hz, 1H, CH=), 5.24 (t, *J* = 7.1 Hz, 1H, CH=), 5.12–5.04 (m, 6H, 6 × CH=), 4.75 (d, *J* = 6.8 Hz, 2H, NCH_2_), 4.39 (d, *J* = 7.3 Hz, 2H, NCH_2_), 2.17–1.94 (m, 24H, 12 × CH_2_), 1.83 (s, 3H, CH_3_), 1.75 (s, 3H, CH_3_), 1.67 (s, 6H, 2 × CH_3_), 1.60 (s, 9H, 3 × CH_3_), 1.59 (s, 6H, 2 × CH_3_), 1.58 (s, 3H, CH_3_); ^13^C{^1^H} NMR (150 MHz, CDCl_3_): δ 172.8 (HC=O), 159.8 (C, C-4), 150.5 (C, C-2), 144.3 (C=), 141.3 (C=), 140.8 (C, C-6), 136.1 (C=), 135.4 (C=), 135.3 (C=), 135.1 (C=), 131.44 (C=), 131.40 (C=), 124.53 (CH=), 124.48 (CH=), 124.4 (CH=), 124.1 (CH=), 124.0 (CH=), 123.4 (CH=), 117.4 (CH=), 116.9 (CH=), 97.4 (C, C-5), 46.6 (NCH_2_), 42.0 (NCH_2_), 39.87 (2 × CH_2_), 39.86 (CH_2_), 39.84 (CH_2_), 39.82 (CH_2_), 39.7 (CH_2_), 26.91 (CH_2_), 26.90 (CH_2_), 26.8 (CH_2_), 26.7 (CH_2_), 26.6 (CH_2_), 26.3 (CH_2_), 25.9 (2 × CH_3_), 17.8 (2 × CH_3_), 16.8 (CH_3_), 16.7 (CH_3_), 16.24 (CH_3_), 16.18 (CH_3_), 16.17 (CH_3_), 16.16 (CH_3_); IR (film from CH_2_Cl_2_): ν_max_ 2964, 2916, 2852, 1622, 1537, 1452 cm^−1^; HRESIMS *m*/*z*: [M + H]^+^ Calcd. for C_45_H_70_N_3_O_2_ 684.5463; Found 684.5487 (Δ = 3.5 ppm); HRESIMS/MS (20 eV) *m*/*z* (%): 684.2009 (2), 412.2985 (13), 140.0471 (100).

#### 3.2.35. (*E*)-2-(3,7-Dimethylocta-2,6-dien-1-yl)isoindoline-1,3-dione (**67**)

Potassium phthalimide (2.6 mmol, 472.8 mg), K_2_CO_3_/Na_2_CO_3_ (1:1, 240 mg), and geranyl bromide (2.5 mmol, 543 mg) in DMF (10 mL) for 22 h yielded **67**, additionally recrystallised from PE after chromatography, 170.2 mg (24%), white crystals. R*_f_* = 0.59 (1:4 EA/PE); ^1^H and ^13^C NMR, and IR data previously reported [56]; HRESIMS *m*/*z*: [M + H]^+^ Calcd. for C_18_H_22_NO_2_ 284.1645; Found 284.1640 (Δ = −1.8 ppm); HRESIMS/MS (40 eV) *m*/*z* (%): 160.0385 (100), 133.0287 (47), 81.0697 (22).

#### 3.2.36. 2-((6*E*)-3,7,11-Trimethyldodeca-2,6,10-trien-1-yl)isoindoline-1,3-dione (**68**)

Potassium phthalimide (2.9 mmol, 541.2 mg), K_2_CO_3_/Na_2_CO_3_ (1:1, 1.9 g) and farnesyl bromide (2.0 mmol, 570.5 mg) in DMF (10 mL) for 19 h yielded **68**, 12.8 mg (2%), colourless oil. R*_f_* = 0.23 (1:10 EA/PE); 5:4 *E*/*Z*, ^1^H and ^13^C NMR data for the (2*E*)-isomer previously reported [57]; NMR data for (2*Z*)-isomer: ^1^H NMR (500 MHz, CDCl_3_): δ 7.84–7.81 (m, 2H, H-5), 7.71–7.68 (m, 2H, H-6), 5.30–5.24 (m, 1H, CH=), 5.11–5.01 (m, 2H, 2 × CH=), 4.27 (d, *J* = 7.1 Hz, 2H, NCH_2_), 2.30–2.24 (m, 2H, CH_2_), 2.01–2.03 (m, 2H, CH_2_), 2.03–1.97 (m, 2H, CH_2_), 1.94–1.87 (m, 2H, CH_2_), 1.82 (s, 3H, CH_3_), 1.63 (s, 3H, CH_3_), 1.56 (s, 6H, 2 × CH_3_); ^13^C{^1^H} NMR (150 MHz, CDCl_3_): δ 168.3 (C, C-1), 140.8 (C=), 135.5 (C=), 133.9 (CH, C-6), 132.5 (C, C-4), 131.7 (C=), 124.6 (CH=), 124.4 (CH=), 123.3 (CH, C-5), 118.1 (CH=), 39.9 (CH_2_), 32.1 (CH_2_), 35.9 (NCH_2_), 26.7 (CH_2_), 26.2 (CH_2_), 25.9 (CH_3_), 23.5 (CH_3_), 17.8 (CH_3_), 16.1 (CH_3_); IR (film from CH_2_Cl_2_): ν_max_ 2964, 2917, 2854, 1710 cm^−1^; HRESIMS *m*/*z*: [M + H]^+^ Calcd. for C_23_H_30_NO_2_ 352.2271; Found 352.2257 (Δ = −4.0 ppm); HRESIMS/MS (40 eV) *m*/*z* (%): 250.9685 (16), 160.0391(100).

#### 3.2.37. 2-((2*E*,6*E*,10*E*)-3,7,11,15-Tetramethylhexadeca-2,6,10,14-tetraen-1-yl)isoindoline-1,3-dione (**69**)

Following the general alkylation procedure, also previously published [58], potassium phthalimide (0.30 mmol, 54.8 mg), K_2_CO_3_ (1.1 mmol, 155.6 mg) and geranylgeranyl bromide (0.32 mmol, 111 mg) yielded **69**, 42.6 mg (45%), colourless oil. R*_f_* = 0.31 (1:9 EA/PE); ^1^H NMR (500 MHz, CDCl_3_): δ 7.83 (dd, *J* = 5.3, 3.0 Hz, 1H, H-5), 7.69 (dd, *J* = 5.4, 3.0 Hz, 1H, H-6), 5.27 (t, *J* = 7.2 Hz, 1H, CH=), 5.13–5.02 (m, 3H, 3 × CH=), 4.28 (d, *J* = 7.1 Hz, 2H, NCH_2_), 2.11–1.89 (m, 12H, 6 × CH_2_), 1.83 (s, 3H, CH_3_), 1.67 (s, 3H, CH_3_), 1.59 (s, 3H, CH_3_), 1.57 (s, 6H, 2 × CH_3_); ^13^C{^1^H} NMR (150 MHz, CDCl_3_): δ 168.2 (C, C-1), 140.8 (C=), 135.4 (C=), 134.9 (C=), 133.9 (CH, C-6), 132.4 (C, C-4), 131.3 (C=), 124.5 (CH=), 124.3 (CH=), 123.7 (CH=), 123.2 (CH, C-5), 118.0 (CH=), 39.8 (CH_2_), 39.7 (CH_2_), 39.6 (CH_2_), 35.9 (NCH_2_), 26.8 (CH_2_), 26.7 (CH_2_), 26.3 (CH_2_), 25.8 (CH_3_), 17.8 (CH_3_), 16.5 (CH_3_), 16.10 (CH_3_), 16.07 (CH_3_); IR (film from CH_2_Cl_2_): ν_max_ 3057, 2930, 1710 cm^−1^; HRESIMS *m*/*z*: [M + H]^+^ Calcd. for C_28_H_38_NO_2_ 420.2897; Found 420.2889 (Δ = −1.9 ppm).

#### 3.2.38. (*E*)-1-(3,7-Dimethylocta-2,6-dien-1-yl)-1*H*-imidazole (**70**)

Following the general alkylation procedure, also previously published [59], imidazole (0.57 mmol, 38.7 mg), K_2_CO_3_ (0.66 mmol, 91.4 mg) and geranyl bromide (0.60 mmol, 130 mg) in DMF (2 mL) for 46 h yielded **70**, 36.7 mg (32%), colourless oil. R*_f_* = 0.20 (1:1 EA/PE); ^1^H NMR data reported previously [60]; ^13^C{^1^H} NMR (120 MHz, CDCl_3_): δ 141.7 (C=), 136.8 (CH, C-2), 132.2 (C=), 129.3 (CH, C-4), 123.6 (CH=), 118.7 (CH, C-5), 118.6 (CH=), 44.6 (NCH_2_), 39.4 (CH_2_), 26.2 (CH_2_), 25.8 (CH_3_), 17.8 (CH_3_), 16.3 (CH_3_); IR (film from CH_2_Cl_2_): ν_max_ 3110, 2966, 2916, 2855 cm^−1^; HRESIMS *m*/*z*: [M + H]^+^ Calcd. for C_13_H_21_N_2_ 205.1699; Found 205.1700 (Δ = 0.5 ppm); HRESIMS/MS (40 eV) *m*/*z* (%): 81.0716 (21), 79.0559 (18), 69.0470 (100).

#### 3.2.39. 1-((6*E*)-3,7,11-Trimethyldodeca-2,6,10-trien-1-yl)-1*H*-imidazole (**71**)

Following the general alkylation procedure, also previously published [59], imidazole (0.51 mmol, 34.6 mg), K_2_CO_3_ (0.65 mmol, 90.4 mg) and farnesyl bromide (0.60 mmol, 171 mg) in DMF (2 mL) for 48 h yielded **71**, 11.4 mg (8%), colourless oil. R*_f_* = 0.10 (1:1 EA/PE); 7:2 *E*/*Z*, NMR data for major isomer: ^1^H NMR (600 MHz, CDCl_3_): δ 7.46 (s, 1H, H-2), 7.04 (s, 1H, H-4), 6.88 (s, 1H, H-5), 5.38–5.33 (m, 1H, CH=), 5.12–5.05 (m, 2H, 2 × CH=), 4.52 (d, *J* = 7.1 Hz, 2H, NCH_2_), 2.19–1.94 (m, 8H, 4 × CH_2_), 1.75 (s, 3H, CH_3_), 1.67 (s, 3H, CH_3_), 1.60 (s, 3H, CH_3_), 1.59 (s, 3H, CH_3_); ^13^C{^1^H} NMR (150 MHz, CDCl_3_): δ 141.9 (C=) 136.8 (CH, C-2), 135.9 (C=), 131.5 (C=), 129.2 (CH, C-4), 124.4 (CH=), 123.5 (CH=), 118.8 (CH, C-5), 118.6 (CH=), 44.7 (NCH_2_), 39.8 (CH_2_), 39.5 (CH_2_), 26.8 (CH_2_), 26.3 (CH_2_), 25.8 (CH_3_), 17.8 (CH_3_), 16.5 (CH_3_), 16.2 (CH_3_); IR (film from CH_2_Cl_2_): ν_max_ 2965, 2917, 2856 cm^−1^; HRESIMS *m*/*z*: [M + H]^+^ Calcd. for C_18_H_29_N_2_ 273.2325; Found 273.2326 (Δ = 0.4 ppm); HRESIMS/MS (20 eV) *m*/*z* (%): 81.0700 (45), 69.0457 (100).

#### 3.2.40. 1-((2*E*,6*E*,10*E*)-3,7,11,15-Tetramethylhexadeca-2,6,10,14-tetraen-1-yl)-1*H*-imidazole (**72**)

Imidazole (0.30 mmol, 20.4 mg), K_2_CO_3_ (0.34 mmol, 46.4 mg) and geranylgeranyl bromide (0.32 mmol, 111 mg) in DMF (1 mL) for 48 h yielded **72**, 16.5 mg (20%), colourless oil. R*_f_* = 0.18 (2:1 EA/PE); ^1^H NMR (500 MHz, CDCl_3_): δ 7.47 (s, 1H, H-2), 7.05 (s, 1H, H-4), 6.89 (s, 1H, H-5), 5.36 (t, *J* = 6.7 Hz, 1H, CH=), 5.12–5.05 (m, 3H, 3 × CH=), 4.52 (d, *J* = 7.1 Hz, 2H, NCH_2_), 2.17–2.01 (m, 8H, 4 × CH_2_), 2.01–1.92 (m, 4H, 2 × CH_2_), 1.75 (s, 3H, CH_3_), 1.67 (s, 3H, CH_3_), 1.59 (s, 9H, 3 × CH_3_); ^13^C{^1^H} NMR (150 MHz, CDCl_3_): δ 142.0 (C=), 136.8 (CH, C-2), 135.9 (C=), 135.1 (C=), 131.4 (C=), 129.1 (CH, C-4), 124.5 (CH=), 124.2 (CH=), 123.5 (CH=), 118.8 (CH, C-5), 118.5 (CH=), 44.7 (NCH_2_), 39.9 (CH_2_), 39.8 (CH_2_), 39.5 (CH_2_), 26.9 (CH_2_), 26.7 (CH_2_), 26.3 (CH_2_), 25.8 (CH_3_), 17.8 (CH_3_), 16.5 (CH_3_), 16.2 (CH_3_), 16.1 (CH_3_); IR (film from CH_2_Cl_2_): ν_max_ 3118, 2966, 2925 cm^−1^; HRESIMS *m*/*z*: [M + H]^+^ Calcd. for C_23_H_37_N_2_ 341.2951; Found 341.2934 (Δ = −5.0 ppm); HRESIMS/MS (40 eV) *m*/*z* (%): 121.1001 (95), 109.1006 (58), 107.0849 (100).

### 3.3. General Formylation Procedure for Synthesis of ***73–77***

#### 3.3.1. (*E*)-*N*-(3,7-Dimethylocta-2,6-dien-1-yl)formamide (**73**)

Acetic anhydride (3.6 mmol, 0.336 g, 339 μL) was stirred on an ice bath for 10 min under nitrogen before addition of formic acid (3.9 mmol, 180 mg, 147.6 μL). The ice bath was removed, and the reaction was heated at 55 °C in an oil bath for 2.5 h. The reaction was cooled to RT and geranylamine (1.6 mmol, 250 mg) was added and reaction was stirred for 3 h. H_2_O (3 mL) was added and the mixture was extracted with EA (3 × 2 mL). The combined extracts were washed with H_2_O (3 × 2 mL), then brine (1 × 2 mL) and dried over anhydrous MgSO_4_. The solvent was removed under reduced pressure and the residue was purified by silica gel flash chromatography (2:3 EA/PE, R*_f_* = 0.21) to yield **73**, 43.3 mg (30%), pale-yellow oil. ^1^H and ^13^C NMR data previously reported [61]; IR (film from CH_2_Cl_2_): ν_max_ 3277, 3045, 2968, 2915, 2856, 1655 cm^−1^; HRESIMS *m*/*z*: [M + H]^+^ Calcd. for C_11_H_20_NO 182.1539; Found 182.1528 (Δ = −6.0 ppm); HRESIMS/MS (20 eV) *m*/*z* (%): 182.1502 (7), 137.1316 (100), 109.0999 (15).

#### 3.3.2. *N*-((2*Z*,6*E*)-3,7,11-Trimethyldodeca-2,6,10-trien-1-yl)formamide (**74**) and *N*-((6*E*)-3,7,11-trimethyldodeca-2,6,10-trien-1-yl)formamide (**75**)

Acetic anhydride (2.5 mmol, 254 mg, 235 μL), formic acid (2.7 mmol, 124 mg, 102 μL), and farnesylamine (1.13 mmol, 250 mg) for 5 h yielded **74** and **75**.

Compound **74**: 3.6 mg (3%), colourless oil; R*_f_* = 0.18 (1:2 EA/PE); 3:1 rotamers, IR, ^1^H and ^13^C NMR data for the (2*Z*)-isomer previously reported [62]; IR (film from CH_2_Cl_2_): ν_max_ 3024, 2964, 2919, 2855, 1656 cm^−1^; HRESIMS *m*/*z*: [M + H]^+^ Calcd. for C_16_H_28_NO 250.2165; Found 250.2161 (Δ = −1.6 ppm); HRESIMS/MS (40 eV) *m*/*z* (%): 109.1014 (74), 107.0841 (100).

Compound **75**: 12.2 mg (7%), colourless oil; R*_f_* = 0.14 (1:2 EA/PE); 3:2 *E*/*Z*, 7:2 rotamers. ^1^H and ^13^C NMR data for the (2*E*)-isomer [63] and (2*Z*)-isomer [62] previously reported; IR (film from CH_2_Cl_2_): ν_max_ 3279, 2965, 2916, 2854, 1657 cm^−1^; HRESIMS *m*/*z*: [M + H]^+^ Calcd. for C_16_H_28_NO 250.2165; Found 250.2159 (Δ = −2.4 ppm); HRESIMS/MS (40 eV) *m*/*z* (%): 121.0998 (60), 109.1014 (100).

#### 3.3.3. *N*-((2*Z*,6*E*,10*E*)-3,7,11,15-Tetramethylhexadeca-2,6,10,14-tetraen-1-yl)formamide (**76**) and *N*-((2*E*,6*E*,10*E*)-3,7,11,15-tetramethylhexadeca-2,6,10,14-tetraen-1-yl)formamide (**77**)

Acetic anhydride (2.3 mmol, 233 mg, 215.5 μL), formic acid (2.5 mmol, 115 mg, 94 μL), and geranylgeranylamine (0.86 mmol, 250 mg) for 5 h yielded **76** and **77**.

Compound **76**: 3.8 mg (6%), pale-yellow oil; R*_f_* = 0.26 (2:3 EA/PE); 2:1 rotamers, data for major rotamer: ^1^H NMR (300 MHz, CDCl_3_): δ 8.14 (s, 1H, HC=O), 5.28 (br s, 1H, NH), 5.21 (t, *J* = 8.4 Hz, 1H, CH=), 5.10 (br s, 3H, 3 × CH=), 3.88 (t, *J* = 6.1 Hz, 2H, NCH_2_), 2.15–1.90 (m, 12H, 6 × CH_2_), 1.74 (s, 3H, CH_3_), 1.68 (s, 6H, 2 × CH_3_), 1.60 (s, 6H, 2 × CH_3_); ^13^C{^1^H} NMR (150 MHz, CDCl_3_): δ 160.9 (HC=O), 141.0 (C=), 136.1 (C=), 135.3 (C=), 131.5 (C=), 124.5 (CH=), 124.2 (CH=), 123.6 (CH=), 120.2 (CH=), 39.9 (2 × CH_2_), 36.0 (NCH_2_), 32.1 (CH_2_), 26.9 (CH_2_), 26.7 (CH_2_), 26.5 (CH_2_), 25.9 (CH_3_), 23.5 (CH_3_), 17.8 (CH_3_), 16.17 (CH_3_), 16.15 (CH_3_); IR (film from CH_2_Cl_2_): ν_max_ 3299, 3024, 2964, 2920, 2874, 1656 cm^−1^; HRESIMS *m*/*z*: [M + H]^+^ Calcd. for C_21_H_36_NO 318.2791; Found 318.2780 (Δ = −3.5 ppm); HRESIMS/MS (20 eV) *m*/*z* (%): 177.1655 (22), 107.0838 (100).

Compound **77**: 6.5 mg (3%), pale-yellow oil; R*_f_* = 0.21 (2:3 EA/PE); 3:1 rotamers, NMR data for major rotamer: ^1^H NMR (300 MHz, CDCl_3_): δ 8.16 (s, 1H, HC=O), 5.41 (br s, 1H, NH), 5.20 (t, *J* = 6.4 Hz, 1H, CH=), 5.10 (br s, 3H, 3 × CH=), 3.90 (t, *J* = 6.1 Hz, 3H, NCH_2_), 2.16–1.90 (m, 12H, 6 × CH_2_), 1.68 (s, 6H, 2 × CH_3_), 1.60 (s, 9H, 3 × CH_3_); ^13^C{^1^H} NMR (150 MHz, CDCl_3_): δ 161.0 (HC=O), 140.8 (C=), 135.7 (C=), 135.2 (C=), 131.5 (C=), 124.5 (CH=), 124.3 (CH=), 123.8 (CH=), 119.4 (CH=), 39.9 (CH_2_), 39.8 (CH_2_), 39.6 (CH_2_), 36.2 (NCH_2_), 26.9 (CH_2_), 26.8 (CH_2_), 26.4 (CH_2_), 25.9 (CH_3_), 17.8 (CH_3_), 16.5 (CH_3_), 16.18 (CH_3_), 16.16 (CH_3_); IR (film from CH_2_Cl_2_): ν_max_ 3293, 2966, 2915, 2853, 1660 cm^−1^; HRESIMS *m*/*z*: [M + H]^+^ Calcd. for C_21_H_36_NO 318.2791; Found 318.2784 (Δ = −2.2 ppm); HRESIMS/MS (20 eV) *m*/*z* (%): 121.0997 (59), 109.0998 (82), 107.0846 (100).

### 3.4. General Acetylation Procedure for Synthesis of ***81–85***

#### 3.4.1. (*E*)-*N*-(3,7-Dimethylocta-2,6-dien-1-yl)acetamide (**81**)

Triethylamine (4.9 mmol, 497 mg, 681.5 μL) and geranylamine (1.6 mmol, 250.1 mg) in THF (15 mL) were stirred on an ice bath for 10 min. Acetyl chloride (2.5 mmol, 0.192 g, 174 μL) was added and the reaction was stirred for 5 h while allowed to warm to room temperature. The reaction was quenched with ice-cold H_2_O (10 mL) and the mixture was extracted with EA (2 × 10 mL). The combined extracts were washed with H_2_O (2 × 10 mL), then brine (1 × 10 mL) and dried over anhydrous MgSO_4_. The solvent was removed under reduced pressure and the resulting residue was purified by silica gel flash chromatography (1:1 EA/PE, R*_f_* = 0.16) to yield **81**, 96.1 mg (30%), pale-yellow oil. ^1^H and ^13^C NMR data previously reported [39]; IR (film from CH_2_Cl_2_): ν_max_ 3293, 3081, 2970, 2928, 1649 cm^−1^; HRESIMS *m*/*z*: [M + H]^+^ Calcd. for C_12_H_22_NO 196.1696; Found 196.1708 (Δ = 6.1 ppm); HRESIMS/MS (40 eV) *m*/*z* (%): 137.1312 (34), 109.0997 (79), 107.0473 (100).

#### 3.4.2. *N*-((2*Z*,6*E*)-3,7,11-Trimethyldodeca-2,6,10-trien-1-yl)acetamide (82) and *N*-((2*E*,6*E*)-3,7,11-trimethyldodeca-2,6,10-trien-1-yl)acetamide (**83**)

Triethylamine (3.4 mmol, 342.7 mg, 472 μL), farnesylamine (1.1 mmol, 250.1 mg), and acetyl chloride (1.7 mmol, 133 mg, 120 μL) yielded **82** and **83**.

Compound **82**: 9.9 mg (9%), pale-yellow oil; R*_f_* = 0.30 (1:1 EA/PE); ^1^H NMR (500 MHz, CDCl_3_): δ 5.32 (br s, 1H, NH), 5.20 (t, *J* = 7.2 Hz, 1H, CH=), 5.12–5.04 (m, 2H, 2 × CH=), 3.82 (t, *J* = 6.5 Hz, 2H, NCH_2_), 2.11–1.97 (m, 8H, 4 × CH_2_), 1.96 (s, 3H, CH_3_C=O), 1.72 (s, 3H, CH_3_), 1.68 (s, 3H, CH_3_), 1.60 (s, 6H, 2 × CH_3_); ^13^C{^1^H} NMR (150 MHz, CDCl_3_): δ 169.9 (C=O), 140.5 (C=), 135.9 (C=), 131.6 (C=), 124.3 (CH=), 123.6 (CH=), 120.7 (CH=), 39.9 (CH_2_), 37.5 (NCH_2_), 32.03 (CH_2_), 26.8 (CH_2_), 26.5 (CH_2_), 25.9 (CH_3_), 23.5 (CH_3_C=O), 23.4 (CH_3_), 17.9 (CH_3_), 16.2 (CH_3_); IR (film from CH_2_Cl_2_): ν_max_ 3293, 3086, 2967, 2927, 2857, 1653 cm^−1^; HRESIMS *m*/*z*: [M + H]^+^ Calcd. for C_17_H_30_NO 264.2322; Found 264.2319 (Δ = −1.1 ppm); HRESIMS/MS (40 eV) *m*/*z* (%): 109.1008 (95), 107.0850 (100).

Compound **83**: 65.1 mg (37%), pale-yellow oil; R*_f_* = 0.22 (1:1 EA/PE); ^1^H NMR data previously reported [41]; ^13^C{^1^H} NMR (150 MHz, CDCl_3_): δ 169.9 (C=O), 140.3 (C=), 135.6 (C=), 131.5 (C=), 124.4 (CH=), 123.8 (CH=), 119.9 (CH=), 39.8 (CH_2_), 39.6 (CH_2_), 37.8 (NCH_2_), 26.9 (CH_2_), 26.4 (CH_2_), 25.9 (CH_3_), 23.4 (CH_3_C=O), 17.8 (CH_3_), 16.4 (CH_3_), 16.2 (CH_3_); IR (film from CH_2_Cl_2_): ν_max_ 3279, 3080, 2966, 2917, 2855, 1659 cm^−1^; HRESIMS *m*/*z*: [M + H]^+^ Calcd. for C_17_H_30_NO 264.2322; Found 264.2318 (Δ = −1.5 ppm); HRESIMS/MS (40 eV) *m*/*z* (%): 109.1000 (57), 107.0843 (100).

#### 3.4.3. *N*-((2*Z*,6*E*,10*E*)-3,7,11,15-Tetramethylhexadeca-2,6,10,14-tetraen-1-yl)acetamide (**84**) and *N*-((2*E*,6*E*,10*E*)-3,7,11,15-tetramethylhexadeca-2,6,10,14-tetraen-1-yl)acetamide (**85**)

Triethylamine (3.16 mmol, 319 mg, 440 μL), geranylgeranylamine (1.0 mmol, 300 mg), and acetyl chloride (1.1 mmol, 89 mg, 81 μL) in THF (10 mL) yielded **84** and **85**.

Compound **84**: 22.1 mg (27%), pale-yellow oil; R*_f_* = 0.20 (1:1 EA/PE); ^1^H NMR (300 MHz, CDCl_3_): δ 5.38 (br s, 1H, NH), 5.19 (t, *J* = 7.2 Hz, 1H, CH=), 5.14–5.03 (m, 3H, 3 × CH=), 3.81 (t, *J* = 6.1 Hz, 2H, NCH_2_), 2.13–1.96 (m, 12H, 6 × CH_2_), 1.95 (s, 3H, CH_3_C=O), 1.72 (s, 3H, CH_3_), 1.67 (s, 3H, CH_3_), 1.59 (s, 9H, 3 × CH_3_); ^13^C{^1^H} NMR (150 MHz, CDCl_3_): δ 169.9 (C=O), 140.4 (C=), 136.0 (C=), 135.2 (C=), 131.4 (C=), 124.5 (CH=), 124.2 (CH=), 123.6 (CH=), 120.7 (CH=), 39.9 (CH_2_), 39.8 (CH_2_), 37.5 (NCH_2_), 32.1 (CH_2_), 26.9 (CH_2_), 26.7 (CH_2_), 26.5 (CH_2_), 25.8 (CH_3_), 23.5 (CH_3_), 23.4 (CH_3_C=O), 17.8 (CH_3_), 16.2 (CH_3_), 16.1 (CH_3_); IR (film from CH_2_Cl_2_): ν_max_ 3277, 3078, 2965, 2916, 2855, 1649 cm^−1^; HRESIMS *m*/*z*: [M + H]^+^ Calcd. for C_22_H_38_NO 332.2948; Found 332.2951 (Δ = 0.9 ppm); HRESIMS/MS (10 eV) *m*/*z* (%): 332.2946 (100), 273.2567 (9).

Compound **85**: 95.5 mg (38%), pale-yellow oil; R*_f_* = 0.14 (1:1 EA/PE); ^1^H NMR (300 MHz, CDCl_3_): δ 5.32 (br s, 1H, NH), 5.19 (t, *J* = 6.8 Hz, 1H, CH=), 5.15–5.04 (m, 3H, 3 × CH=), 3.84 (t, *J* = 6.1 Hz, 2H, NCH_2_), 2.15–1.98 (m, 12H, 6 × CH_2_), 1.67 (s, 6H, 2 × CH_3_), 1.60 (s, 9H, 3 × CH_3_); ^13^C{^1^H} NMR (150 MHz, CDCl_3_): δ 169.9 (C=O), 140.3 (C=), 135.6 (C=), 135.2 (C=), 131.5 (C=), 124.5 (CH=), 124.3 (CH=), 123.8 (CH=), 119.9 (CH=), 39.9 (CH_2_), 39.8 (CH_2_), 39.6 (CH_2_), 37.8 (NCH_2_), 26.9 (CH_2_), 26.8 (CH_2_), 26.5 (CH_2_), 25.9 (CH_3_), 23.5 (CH_3_C=O), 17.8 (CH_3_), 16.5 (CH_3_), 16.19 (CH_3_), 16.17 (CH_3_); IR (film from CH_2_Cl_2_): ν_max_ 3279, 3078, 2967, 2917, 2854, 1650 cm^−1^; HRESIMS *m*/*z*: [M + H]^+^ Calcd. for C_22_H_38_NO 332.2948; Found 332.2934 (Δ = −4.2 ppm); HRESIMS/MS (20 eV) *m*/*z* (%): 332.3002 (100), 149.1318 (100).

### 3.5. General Methylation Procedure for Synthesis of ***78–80*** and ***86–88***

#### 3.5.1. (*E*)-*N*-(3,7-Dimethylocta-2,6-dien-1-yl)-*N*-methylformamide (**78**)

Freshly powdered KOH (0.647 mmol, 36.3 mg) was stirred in dry DMSO (0.1 mL) for 10 min before the addition of **73** (0.17 mmol, 30.0 mg, in 0.1 mL DMSO), followed immediately by the addition of methyl iodide (0.5 mmol, 71.0 mg, 31.2 µL). The reaction was stirred for 2.5 h, then poured onto H_2_O (3 mL) and extracted with DCM (3 × 2 mL). The combined extracts were washed with H_2_O (5 × 2 mL), then brine (1 × 2 mL) and dried over anhydrous MgSO_4_. The solvent was removed under reduced pressure and the resulting residue was purified by silica gel flash chromatography (1:3 EA/PE, R*_f_* = 0.14) to yield **78**, 17.4 mg (54%), colourless oil. 3:2 rotamers, data for major rotamer: ^1^H NMR (500 MHz, CDCl_3_): δ 8.06 (s, 1H, HC=O), 5.11–5.06 (m, 1H, CH=), 5.06–5.01 (m, 1H, CH=), 3.80 (d, *J* = 7.1 Hz, 2H, NCH_2_), 2.78 (s, 3H, NCH_3_), 2.13–1.99 (m, 4H, 2 × CH_2_), 1.67 (s, 3H, CH_3_), 1.66 (s, 3H, CH_3_), 1.58 (s, 3H, CH_3_); ^13^C{^1^H} NMR (150 MHz, CDCl_3_): δ 162.5 (HC=O), 141.3 (C=), 132.1 (C=), 123.7 (CH=), 119.1 (CH=), 47.2 (NCH_2_), 39.7 (CH_2_), 29.2 (NCH_3_), 26.3 (CH_2_), 25.8 (CH_3_), 17.8 (CH_3_), 16.30 (CH_3_); IR (film from CH_2_Cl_2_): ν_max_ 2965, 2917, 2854, 1663 cm^−1^; HRESIMS *m*/*z*: [M + H]^+^ Calcd. for C_12_H_22_NO 196.1696; Found 196.1695 (Δ = −0.5 ppm); HRESIMS/MS (20 eV) *m*/*z* (%): 196.1694 (17), 137.1319 (100), 109.1010 (21).

#### 3.5.2. *N*-Methyl-*N*-((6*E*)-3,7,11-trimethyldodeca-2,6,10-trien-1-yl)formamide (**79**)

KOH (0.18 mmol, 9.9 mg), **75** (0.040 mmol, 10 mg), and methyl iodide (0.12 mmol, 17 mg, 7.5 µL) yielded **79**, 5.9 mg (56%), colourless oil. R*_f_* = 0.10 (1:4 EA/PE); 3:2 *E*/*Z*, 3:2 rotamers, data for major isomer and rotamer: ^1^H NMR (500 MHz, CDCl_3_): δ 8.08 (s, 1H, CH=O), 5.13–5.04 (m, 3H, 3 × CH=), 3.81 (d, *J* = 7.0 Hz, 2H, NCH_2_), 2.79 (s, 3H, NCH_3_), 2.14–2.00 (m, 6H, 3 × CH_2_), 1.99–1.93 (m, 2H, CH_2_), 1.68 (s, 3H, CH_3_), 1.64 (s, 3H, CH_3_), 1.59 (s, 6H, 2 × CH_3_); ^13^C{^1^H} NMR (150 MHz, CDCl_3_): δ 162.6 (HC=O), 141.4 (C=), 135.8 (C=), 131.5 (C=), 124.4 (CH=), 123.6 (CH=), 119.0 (CH=), 47.3 (NCH_2_), 39.9 (CH_2_), 39.7 (CH_2_), 29.2 (NCH_3_), 26.8 (CH_2_), 26.3 (CH_2_), 25.9 (CH_3_), 17.8 (CH_3_), 16.4 (CH_3_), 16.2 (CH_3_); IR (film from CH_2_Cl_2_): ν_max_ 3495, 2963, 2916, 2853, 1677, 1665 cm^−1^; HRESIMS *m*/*z*: [M + H]^+^ Calcd. for C_17_H_30_NO 264.2322; Found 264.2324 (Δ = 0.8 ppm); HRESIMS/MS (40 eV) *m*/*z* (%): 109.1001 (53), 107.085 (100), 105.0694 (58).

#### 3.5.3. *N*-Methyl-*N*-((2*E*,6*E*,10*E*)-3,7,11,15-tetramethylhexadeca-2,6,10,14-tetraen-1-yl)formamide (**80**)

KOH (0.201 mmol, 11.3 mg), **77** (0.047 mmol, 15 mg), and methyl iodide (0.14 mmol, 20 mg, 8.8 µL) yielded **80**, 5.2 mg (33%), colourless oil. R*_f_* = 0.13 (1:3 EA/PE); ^1^H NMR (500 MHz, CDCl_3_): δ 8.08 (s, 1H, HC=O), 5.15–5.03 (m, 4H, 4 × CH=), 3.81 (d, *J* = 7.0 Hz, 2H, NCH_2_), 2.79 (s, 3H, NCH_3_), 2.14–2.01 (m, 8H, 4 × CH_2_), 2.01–1.88 (m, 4H, 2 × CH_2_), 1.69 (s, 3H, CH_3_), 1.68 (s, 3H, CH_3_), 1.61 (s, 3H, CH_3_), 1.60 (s, 6H, 2 × CH_3_); ^13^C{^1^H} NMR (150 MHz, CDCl_3_): δ 162.5 (HC=O), 141.4 (C=), 135.8 (C=), 135.2 (C=), 131.4 (C=), 124.5 (CH=), 124.2 (CH=), 123.6 (CH=), 119.0 (CH=), 47.3 (NCH_2_), 39.9 (CH_2_), 39.8 (CH_2_), 39.7 (CH_2_), 29.2 (NCH_3_), 26.9 (CH_2_), 26.7 (CH_2_), 26.4 (CH_2_), 25.9 (CH_3_), 17.8 (CH_3_), 16.4 (CH_3_), 16.18 (CH_3_), 16.15 (CH_3_); IR (film from CH_2_Cl_2_): ν_max_ 2962, 2922, 2854, 1681 cm^−1^; HRESIMS *m*/*z*: [M + H]^+^ Calcd. for C_22_H_38_NO 332.2948; Found 322.2955 (Δ = 2.2 ppm); HRESIMS/MS (40 eV) *m*/*z* (%): 123.1153 (39), 121.003 (51), 109.1002 (74), 107.0846 (100).

#### 3.5.4. *N*-Methyl-*N*-((2*E*,6*E*)-3,7,11-trimethyldodeca-2,6,10-trien-1-yl)acetamide (**86**)

KOH (0.18 mmol, 9.9 mg), **83** (0.04 mmol, 10 mg), and methyl iodide (0.12 mmol, 17 mg, 7.5 μL) yielded **86**, 5.9 mg (53%), colourless oil. R*_f_* = 0.06 (1:4 EA/PE); *E*/*Z* 2:1, NMR data major isomer and rotamer: ^1^H NMR (500 MHz, CDCl_3_): δ 5.15–5.06 (m, 3H, 3 × CH=), 4.01 (d, *J* = 7.0 Hz, 2H, NCH_2_), 2.91 (s, 3H, CH_3_C=O), 2.14–2.10 (m, 2H, CH_2_), 2.10 (s, 3H, NCH_3_), 2.07–2.02 (m, 4H, 2 × CH_2_), 2.00–1.95 (m, 2H, CH_2_), 1.69 (s, 3H, CH_3_), 1.68 (s, 3H, CH_3_), 1.61 (s, 6H, 2 × CH_3_); ^13^C{^1^H} NMR (150 MHz, CDCl_3_): δ 170.3 (C=O), 139.6 (C=), 135.7 (C=), 131.5 (C=), 124.4 (CH=), 123.6 (CH=), 119.6 (CH=), 48.7 (NCH_2_), 39.9 (CH_2_), 39.6 (CH_2_), 35.2 (NCH_3_), 26.9 (CH_2_), 26.4 (CH_2_), 25.8 (CH_3_), 21.6 (CH_3_C=O), 17.8 (CH_3_), 16.4 (CH_3_), 16.2 (CH_3_); IR (film from CH_2_Cl_2_): ν_max_ 2964, 2916, 2856, 1646 cm^−1^; HRESIMS *m*/*z*: [M + H]^+^ Calcd. for C_18_H_32_NO 278.2478; Found 278.2482 (Δ = 1.4 ppm); HRESIMS/MS (40 eV) *m*/*z* (%): 121.0993(33), 107.0841 (100).

#### 3.5.5. *N*-Methyl-*N*-((2*Z*,6*E*,10*E*)-3,7,11,15-tetramethylhexadeca-2,6,10,14-tetraen-1-yl)acetamide (**87**)

KOH (0.32 mmol, 18.0 mg), **84** (0.051 mmol, 17.0 mg), and methyl iodide (0.15 mmol, 21.8 mg, 9.6 μL) yielded **87**, 5.1 mg (29%), colourless oil. R*_f_* = 0.04 (1:5 EA/PE); ^1^H NMR (500 MHz, CDCl_3_): δ 5.17–5.07 (m, 4H, 4 × CH=), 4.00 (d, *J* = 7.0 Hz, 2H, NCH_2_), 2.92 (s, 3H, NCH_3_), 2.14–2.10 (m, 4H, 2 × CH_2_), 2.10 (s, 3H, CH_3_C=O), 2.09–2.04 (m, 4H, 2 × CH_2_), 2.03–1.97 (m, 4H, 2 × CH_2_), 1.77 (s, 3H, CH_3_), 1.69 (s, 3H, CH_3_), 1.63 (s, 3H, CH_3_), 1.61 (s, 6H, 2 × CH_3_); ^13^C{^1^H} NMR (150 MHz, CDCl_3_): δ 170.5 (C=O), 139.8 (C=), 136.1 (C=), 135.8 (C=), 131.4 (C=), 124.5 (CH=) 124.3 (CH=), 124.1 (CH=), 120.3 (CH=), 48.5 (NCH_2_), 39.9 (2 × CH_2_), 35.3 (NCH_3_), 32.2 (CH_2_), 26.9 (CH_2_), 26.7 (CH_2_), 26.4 (CH_2_), 25.9 (CH_3_), 23.5 (CH_3_), 22.0 (CH_3_C=O), 17.8 (CH_3_), 16.17 (CH_3_), 16.16 (CH_3_); IR (film from CH_2_Cl_2_): ν_max_ 2965, 2915, 2854, 1650 cm^−1^; HRESIMS *m*/*z*: [M + H]^+^ Calcd. for C_23_H_40_NO 346.3104; Found 346.3113 (Δ = 2.6 ppm); HRESIMS/MS (40 eV) *m*/*z* (%):123.1163 (46), 121.1009 (100).

#### 3.5.6. *N*-Methyl-*N*-((2*E*,6*E*,10*E*)-3,7,11,15-tetramethylhexadeca-2,6,10,14-tetraen-1-yl)acetamide (**88**)

KOH (0.36 mmol, 20.3 mg), **85** (0.090 mmol, 30 mg), and methyl iodide (0.27 mmol, 38.7 mg, 17.0 μL) yielded **88**, 4.7 mg (15%), colourless oil. R*_f_* = 0.05 (1:5 EA/PE); ^1^H NMR (500 MHz, CDCl_3_): δ 5.17–5.07 (m, 4H, 4 × CH=), 4.02 (d, *J* = 7.0 Hz, 2H, NCH_2_), 2.92 (s, 3H, NCH_3_), 2.15–2.11 (m, 4H, 2 × CH_2_), 2.10 (s, 3H, CH_3_C=O), 2.08–2.04 (m, 4H, 2 × CH_2_), 2.01–1.96 (m, 4H, 2 × CH_2_), 1.69 (s, 6H, 2 × CH_3_), 1.64 (s, 3H, CH_3_), 1.61 (s, 6H, 2 × CH_3_); ^13^C{^1^H} NMR (150 MHz, CDCl_3_): δ 170.5 (C=O), 139.7 (C=), 135.7 (C=), 135.2 (C=), 131.4 (C=), 124.5 (CH=), 124.2 (CH=), 123.6 (CH=), 119.4 (CH=), 48.7 (NCH_2_), 39.9 (2 × CH_2_), 39.6 (CH_2_), 35.2 (NCH_3_), 26.9 (2 × CH_2_), 26.4 (CH_2_), 25.9 (CH_3_), 22.0 (CH_3_C=O), 17.8 (CH_3_), 16.4 (CH_3_), 16.19 (CH_3_), 16.15 (CH_3_); IR (film from CH_2_Cl_2_): ν_max_ 2965, 2916, 2854, 1650 cm^−1^; HRESIMS *m*/*z*: [M + H]^+^ Calcd. for C_23_H_40_NO 346.3104; Found 346.3109 (Δ = 1.4 ppm); HRESIMS/MS (40 eV) *m*/*z* (%):123.1161 (47), 107.0851 (100).

### 3.6. Purification of TbHsp70, TbHsp70.4, HsHSPA8 and HsDNAJB2

*Escherichia coli* (*E. coli.*) bacterial cells transformed with the respective expression vector were grown at 37 °C in 2× YT medium supplemented with respective antibiotic and were grown to mid-logarithmic phase (A_600_ 0.4–0.6). Protein production was induced by the addition of 1 mM IPTG (isopropyl-β-d-thiogalactopyranoside), and the bacterial cultures were incubated at 37 °C for 3 h for TbHsp70, HsHSPA8, HsDNAJB2, and 1 h for TbHsp70.4. Bacterial cells expressing the recombinant proteins were harvested by centrifugation (10,000 *g*; 15 min; 4 °C) and the cell pellet was resuspended in lysis buffer (100 mM Tris-HCl, pH 7.5, 300 mM NaCl, 20 mM imidazole, 1 mM PMSF, 1 mg/mL lysozyme), allowed to stand for 30 min at room temperature and then frozen at −80 °C overnight. The cells were then thawed on ice and sonicated at 4 °C. The resulting lysate was cleared by centrifugation (13,000 *g*, 40 min, 4 °C) and the supernatant was incubated with cOmplete His-tag purification resin (Roche, Germany) and allowed to bind overnight at 4 °C with gentle agitation. The resin was then pelleted by centrifugation (4500 *g*; 4 min) to remove unbound proteins and washed three times using native wash buffer (100 mM Tris-HCl, pH 7.5, 300 mM NaCl, 50 mM imidazole, 1 mM PMSF) to remove non-specific contaminants. The bound protein was eluted three times by re-suspending the resin in elution buffer (10 mM Tris-HCl, pH 7.5, 300 mM NaCl, 750 mM imidazole). The eluted proteins were extensively dialyzed using SnakeSkin dialysis tubing (Pierce-MWCO 10,000; Thermo Scientific, Waltham, MA, USA) into the appropriate assay buffer for functional studies and then subsequently concentrated against PEG 20,000 (Merck, Darmstadt, Germany). The protein yield was estimated using the Bradford assay (Sigma-Aldrich, St. Louis, MO, USA) with BSA as the standard. SDS-PAGE (10%) and Western analysis using mouse monoclonal anti-His primary antibody and HRP-conjugated goat anti-mouse IgG secondary antibody (Santa Cruz Biotechnology, Inc., Dallas, TX, USA) were conducted to assess the expression and purification of the recombinant proteins (Appendix A). HRP-conjugated goat anti-rabbit (Santa Cruz Biotechnology, Inc., Dallas, TX, USA) was used as the secondary antibody. Imaging of the protein bands on the blot was conducted using the ECL kit (Thermo Scientific, Waltham, MA, USA) as per the manufacturer’s instructions. Images were captured using the ChemiDoc Imaging system (Bio-Rad, Hercules, CA, USA).

### 3.7. Purification of Tbj2

Recombinant N-terminal His-tagged Tbj2 was expressed and purified under native conditions using nickel affinity chromatography from *E. coli* BL21 (DE3) cells as previously described [64]. Samples were dialyzed into the appropriate assay buffer for functional studies.

### 3.8. Growth and Maintenance of T. b. brucei Cultures

Bloodstream form *T. b. brucei* Lister 927 variant 221 strain parasites were cultured in filter sterilized complete Iscoves Modified Dulbeccos Media (IMDM) based HM1-9 medium (IMDM base powder, 3.6 mM sodium bicarbonate, 1 mM hypoxanthine, 1 mM sodium pyruvate, 0.16 mM thymidine, 0.05 mM bathocuprone sulphate acid, 10% (*v*/*v*) heat inactivated foetal bovine serum, 1.5 mM L-cysteine, 0.2 mM β-mercaptoethanol, pH 7.5) in a humidified chamber at 37 °C with an atmosphere of 5% CO_2_. Parasite growth was monitored using a Neubauer haemocytometer to count the cell number, after which cells were diluted, according to their density, in the described pre-warmed media.

### 3.9. Assessment of the Anti-Trypanosomal Activity of the Compounds on Bloodstream Form T. b. brucei Parasites

All compounds of interest were resuspended to stocks of 30 mM in DMSO and stored at −80 °C prior to use in *in vitro* experiments. For assessment of anti-trypanosomal activity, compounds were added to *in vitro* cultures of bloodstream form *T. b. brucei* parasites (1 × 10^5^ cells/mL) in 96-well plates at a fixed concentration of 20 μM. After an incubation period of 48 h, the number of parasites surviving drug exposure were determined using a resazurin-based cytotoxicity assay [65]. Resazurin is an oxidation–reduction sensitive dye that is reduced by living cells to resorufin. Resorufin is a fluorophore (Excitation_560_/Emission_590_) and thus was quantified in a multi-well fluorescence plate reader. Results are expressed as % parasite viability—the resorufin fluorescence in small molecule-treated wells relative to untreated controls. Assessment of each small molecule was conducted in duplicate, with error bars representing standard deviation (SD). The compounds that displayed 80% parasite growth inhibition when tested at 20 μM were evaluated in a dose–response experiment. The compounds were added to *in vitro* cultures of bloodstream form *T. b. brucei* parasites (1 × 10^5^ cells/mL) in 96-well plates in a 3-fold dilution series with 100 μM as the highest concentration. After an incubation period of 48 h, the number of parasites surviving drug exposure were determined as previously mentioned. The % parasite viability was determined as previously mentioned. The IC_50_ (the concentration of compound required to decrease cell viability of *T. b. brucei* parasites by 50% compared to those grown in the absence of the compound) values for each of the compounds were determined from a dose–response curve by non-linear regression generated using GraphPad Prism^®^ (v. 7.0; San Diego, CA, USA) software. For comparative purposes, pentamidine at a fixed concentration of 1 μM was employed as a drug standard.

### 3.10. Growth and Maintenance of HeLa Cells

HeLa cells were grown and maintained in a culture medium comprised of Dulbecco’s Modified Eagle’s Medium (DMEM) with 5 mM L-glutamine (Lonza, Basel, Switzerland), supplemented with 10% (*v*/*v*) heat inactivated foetal bovine serum and antibiotics (penicillin/streptomycin/fungizone-PSF) in a humidified chamber at 37 °C with an atmosphere of 5% CO_2_. To carry out passaging of cells, cultures were treated with 1% trypsin (*w*/*v*) to lift cells, which were then washed in 1 × phosphate buffered saline (PBS) before being re-seeded into culture flasks.

### 3.11. Assessment of the Cytotoxicity of the Small Molecules on a Mammalian Cell Line

For assessment of the cytotoxic effects, the small molecules were added to HeLa cells at a fixed concentration of 20 μM. After an incubation period of 24 h, the number of cells surviving drug exposure were determined using a resazurin-based cytotoxicity assay. The resazurin-based cytotoxicity assay was conducted as previously mentioned. Results were expressed as % cell—the resorufin fluorescence in small molecule-treated wells relative to untreated controls. Assessment of each small molecule was conducted in duplicate, with error bars representing standard deviation (SD). For comparative purposes, emetine at a fixed concentration of 10 μM was employed as a drug standard.

### 3.12. MDH Aggregation Suppression Assay

The described assay was adapted as a tool for screening the modulatory effects of the selected compounds on the chaperone function of the *T. brucei* Hsp70s. As an initial screen of the modulatory effects of the compounds on the chaperone function of the Hsp70s, compounds were used at a concentration of 300 μM. For compounds identified to have a modulatory effect on the chaperone function of Hsp70, a concentration-dependency experiment was conducted where the screening expanded the concentration range (0, 50, 150, and 300 μM) of the compound. These experiments were conducted on three independently purified batches of proteins, but due to limited availability of the selected compounds, each assay was conducted in duplicate on each batch of protein. Several controls were incorporated into the study. To evaluate the effect of DMSO on the chaperone activity of the Hsp70s, the solvent was added to a final concentration of 1% (*v*/*v*) in the assay with MDH and the *T. brucei* Hsp70. The selected compounds were also assayed (at the maximum concentration: 300 μM) together with MDH (no chaperone) to rule out that the compounds were causing an increase or decrease in MDH aggregation; and together with chaperones (no MDH) to ensure that the modulatory of the chaperone activity was not due to aggregation of the Hsp70 in the presence of compounds. The assays were performed in duplicate on three independently purified batches of proteins.

### 3.13. ATPase Activity Assay

The determination of the ATPase activity of the Hsp70 proteins was performed using the high throughput colorimetric ATPase assay kit (Innova Biosciences, Cambridge, UK). This method enables the quantification of the inorganic phosphate (Pi) released from ATP hydrolysis by an enzyme. Briefly, the molecular chaperones were prepared in ATPase assay buffer (100 mM Tris-HCl, 7.5, 2 mM MgCl_2_, 50 mM KCl, 0.5 mM DTT) and incubated with ATP (0–2 mM) for 1 h at 37 °C. A negative control did not contain the enzyme. The samples containing Pi hydrolysed from ATP were incubated with the PiColorLock™ solution, which is a malachite green dye solution that in the presence of Pi changes absorbance due to generation of molybdate-phosphate complexes. The absorbance was measured at 595 nm using a Powerwave 96-well plate reader (BioTek Instruments, Inc., Winooski, VT, USA), and absorbance values were converted to phosphate concentrations using a standard curve of absorbance vs. phosphate concentration based on a set of Pi standards provided by the supplier assayed along with the samples. All samples were corrected for spontaneous breakdown of ATP observed in a control experiment in the absence of protein. The trypanosomal and human Hsp70s were used at 0.8 μM, and the J-proteins were used at 0.4 μM. An initial screen of the modulatory effects of the compounds on the ATPase activity of the Hsp70s was conducted using 300 μM of the compounds. Compounds identified to have a modulatory effect on the basal and J-protein-stimulated ATPase activity of Hsp70 were used at varying concentrations. The modulatory effect of the selected compounds on the basal and J-protein stimulated ATPase activity of the *T. brucei* Hsp70s was represented as fold change with the basal and J-protein stimulated ATPase activity of the Hsp70s taken as 1 respectively. These experiments were conducted on three independently purified batches of proteins, but due to limited availability of the selected compounds, each assay was conducted in duplicate on each batch of protein. In control reactions to evaluate the effect of DMSO on basal and J-protein stimulated ATPase activity of the Hsp70s, the solvent was added to a final concentration of 1% (*v*/*v*) in the assay. The assays were performed in duplicate on three independently purified batches of proteins.

## 4. Conclusions

In conclusion, a large number of malonganenone and nuttingin analogues have been synthesized and assessed for SAR relating to the anti-trypanosomal activity. Some relevant observations include a general increase in anti-parasitic activity with increasing lipophilicity (side-chain length) and certain head-groups are responsible for heightened activity over others. Of particular note is that the greatest anti-parasitic activity was achieved with compounds **47** and **48**, which share the same head-group but vary in the position of the side chain, indicating flexibility within the SAR to allow for variation around both geometry and position of the lipophilic portion of the drug. In addition, cytotoxicity studies have shown that the compounds tested are generally inactive against mammalian cells, while biochemical studies have shown that these compounds potentially modulate the Hsp70 basal chaperone activity, but not the co-chaperone-stimulated chaperone activity. Overall, our study has therefore opened a new series of drug leads and could support the rational design of more active chemotherapeutic agents in the future. Such work is ongoing in our laboratories.

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
