# Peer review of "Screening for Small Molecule Modulators of Trypanosoma brucei Hsp70 Chaperone Activity Based upon Alcyonarian Coral-Derived Natural Products"

_marinedrugs, 2020, doi:10.3390/md18020081_

Round 1

Reviewer 1 Report

In this paper the authors present data in support of some new approaches to antitrypanosomal therapy. As they point out there is resistance to many existing drugs including pentamidine suggesting that there is a need for new antitrypanosomal drugs. My understanding is that with respect to Human African Trypanosomiasis major advances have been obtained through public health measures and therefore other species of trypanosome such as T. cruzi, which is prevalent in South America and more poorly controlled, might be better targets for the development of new drugs. Perhaps the introduction could be sharpened.

There are several strengths to the paper. A good number of compounds has been evaluated and the most active taken into biological evaluation pointing towards the establishment of new targets for antitrypanosomal drugs. This is valuable and is for me a justification for publication.

I would like to suggest that the authors revise their discussion of SAR for a number of reasons. Firstly, none of the compounds is really strongly active (compare the activity of pentamidine, for example); this diminishes the significances of the differences observed in terms of SAR because more than one biological target is likely to be engaged. Secondly, and related to the first, the SAR is in general rather flat, which makes the identification of important structural differences uncertain. Thirdly, I found the discussion a little subjective. There are several compound with IC50s close to 1 uM but not all were treated equally in discussion. I would substantially reduce the length of this section.

On a detailed point of medicinal chemistry, I don't think that the reference to Lipinski's rules makes a convincing point. These compounds all contain highly lipophilic motifs that would be expected to engage with many biological systems, both membranes and proteins. Many medicinal chemists would say that the characteristics of the compounds presented are actually unsuitable as the basis for drug discovery.

With respect to presentation and to support the discussion I think it is necessary to include the IC50 data in the paper itself. A composite table with structures, compound activities, and reference numbers is easily includable in the main text.

Overall I recommend publication on the basis of the biological concepts that the work opens up but recommend that the SAR and medicinal chemistry section be modified.

Author Response

Assoc Prof Joanne Harvey

Marine Drugs

School of Chemical & Physical Sciences

Victoria University of Wellington

Wellington

NEW ZEALAND

RE: Manuscript submission Marine Drugs 680585 “Screening for small molecules modulators of Trypanosoma brucei Hsp70 chaperone activity based upon alcyonarian coral-derived natural products.

January 15, 2020

Dear Associate Professor Harvey,

Many thanks for providing the referees comments and allowing us time to consider and implement some of their suggestions. Here are our responses to referee 1’s comments:

In this paper the authors present data in support of some new approaches to antitrypanosomal therapy. As they point out there is resistance to many existing drugs including pentamidine suggesting that there is a need for new antitrypanosomal drugs. My understanding is that with respect to Human African Trypanosomiasis major advances have been obtained through public health measures and therefore other species of trypanosome such as T. cruzi, which is prevalent in South America and more poorly controlled, might be better targets for the development of new drugs. Perhaps the introduction could be sharpened.

Response: As we did not work on T. cruzi we have not expanded our introduction to include it.

There are several strengths to the paper. A good number of compounds has been evaluated and the most active taken into biological evaluation pointing towards the establishment of new targets for antitrypanosomal drugs. This is valuable and is for me a justification for publication.

Response: We thank the reviewer for these kind comments.

I would like to suggest that the authors revise their discussion of SAR for a number of reasons. Firstly, none of the compounds is really strongly active (compare the activity of pentamidine, for example); this diminishes the significances of the differences observed in terms of SAR because more than one biological target is likely to be engaged. Secondly, and related to the first, the SAR is in general rather flat, which makes the identification of important structural differences uncertain. Thirdly, I found the discussion a little subjective. There are several compound with IC50s close to 1 uM but not all were treated equally in discussion. I would substantially reduce the length of this section.

Response: Lines mentioning the potency of the most active compounds have been removed. The SAR section has been shortened and combined with the previous section as requested.

On a detailed point of medicinal chemistry, I don't think that the reference to Lipinski's rules makes a convincing point. These compounds all contain highly lipophilic motifs that would be expected to engage with many biological systems, both membranes and proteins. Many medicinal chemists would say that the characteristics of the compounds presented are actually unsuitable as the basis for drug discovery.

Response: All mention of Lipinski’s rules have been deleted.

With respect to presentation and to support the discussion I think it is necessary to include the IC50 data in the paper itself. A composite table with structures, compound activities, and reference numbers is easily includable in the main text.

Response: We have added a table of IC50 data with compound reference numbers in the paper. There is an additional table in the supplementary information with structures, which is too long to include in the main paper and the structures are already available in the manuscript text.

Overall I recommend publication on the basis of the biological concepts that the work opens up but recommend that the SAR and medicinal chemistry section be modified.

Response: Again, we thank the reviewer for these kind comments.

We hope that these changes to our manuscript will be suitable to allow for acceptance of our paper. We look forward to hearing the results of our resubmission soon.

Reviewer 2 Report

Authors synthesized 74 analogues of malonganenone & nuttingin and tested their anti-trypanosomal activities. Although their toxicity towards the parasite is unfortunately way lower than commercial drugs like pentamidine, this study is still valuable in exploration of new drugs to fight drug resistance, and in understanding structure-activity relationship.

Over all, the study was carefully conducted and the manuscript was well written. Two points for improvement:
1. Page 8, it will be better to assess the "structure-activity relationship" in a more systematic way, for example, by plotting activity ~ side-chain length relationship for each head group.
2. Page 7, please explain "based on available mass".

Typo:
1. Page 2, hyphen is needed in "1,4 naphthoquinones".
2. Page 2, the 27 in "27 malonganenones" is possibly a typo. Should it be 17?
3. Figure 1, "Malonganenone L" was duplicated.
4. Scheme 8, compound "78" is likely 68.

Author Response

Dear Associate Professor Harvey,

Many thanks for providing the referees comments and allowing us time to consider and implement some of their suggestions. Here are our responses to referee 2’s comments:

Authors synthesized 74 analogues of malonganenone & nuttingin and tested their anti-trypanosomal activities. Although their toxicity towards the parasite is unfortunately way lower than commercial drugs like pentamidine, this study is still valuable in exploration of new drugs to fight drug resistance, and in understanding structure-activity relationship.

Response: We thank the referee for their kind comments.

Over all, the study was carefully conducted and the manuscript was well written. Two points for improvement: 
1. Page 8, it will be better to assess the "structure-activity relationship" in a more systematic way, for example, by plotting activity ~ side-chain length relationship for each head group.
2. Page 7, please explain "based on available mass".

Response: The first suggestion of this referee is in direct contrast of referee 1 who has suggested a reduction in our SAR analysis, given the very similar IC50 values we have obtained. We have therefore not added additional SAR analysis to the manuscript.

For the second suggestion, these samples had considerably more mass available for further testing, than some of the more active compounds therefore we have changed to “with abundantly available mass”.

Typo:
1. Page 2, hyphen is needed in "1,4 naphthoquinones".
2. Page 2, the 27 in "27 malonganenones" is possibly a typo. Should it be 17?
3. Figure 1, "Malonganenone L" was duplicated.
4. Scheme 8, compound "78" is likely 68.

Response: All typographical errors have been amended.

We hope that these changes to our manuscript will be suitable to allow for acceptance of our paper. We look forward to hearing the results of our resubmission soon.
